# Supercharging Graph Transformers with Advective Diffusion

**Qitian Wu** [1]   **Chenxiao Yang** [2]   **Kaipeng Zeng** [3]   **Michael Bronstein** [4][5]

## Abstract

The capability of generalization is a cornerstone for the success of modern learning systems. For non-Euclidean data, e.g., graphs, that particularly involves topological structures, one important aspect neglected by prior studies is how machine learning models generalize under topological shifts. This paper proposes ADVDIFFORMER, a physics-inspired graph Transformer model designed to address this challenge. The model is derived from advective diffusion equations which describe a class of continuous message passing process with observed and latent topological structures. We show that ADVDIFFORMER has provable capability for controlling generalization error with topological shifts, which in contrast cannot be guaranteed by graph diffusion models, i.e., the generalization of common graph neural networks in continuous space. Empirically, the model demonstrates superiority in various predictive tasks across information networks, molecular screening and protein interactions[1].

## 1. Introduction

Learning representations for non-Euclidean data is essential for geometric deep learning. Graph-structured data in particular has attracted increasing attention, as graphs are a very popular mathematical abstraction for systems of relations and interactions that can be applied from microscopic scales (e.g. molecules) to macroscopic ones (social networks). Common frameworks for learning on graphs include graph neural networks (GNNs) (Scarselli et al., 2008; Gilmer et al., 2017; Kipf & Welling, 2017), which operate by propagating information between adjacent nodes of the graph networks,

and graph Transformers (Ying et al., 2021; Wu et al., 2021; Rampásek et al., 2022; Wu et al., 2022b), which propagate information among arbitrary node pairs through global attention. GNNs can be seen as discretized versions of local diffusion equations on graphs (Atwood & Towsley, 2016; Klicpera et al., 2019; Chamberlain et al., 2021a), while graph Transformers can be considered as the counterparts of non-local diffusion (Wu et al., 2023; 2024c). Linking graph learning models with diffusion equations offers powerful tools from the domain of partial differential equations (PDEs), allowing us to study the expressive power (Bodnar et al., 2022), behaviors such as over-smoothing (Rusch et al., 2023) and over-squashing (Topping et al., 2022), the settings of missing features (Rossi et al., 2022), and guide architectural choices (Di Giovanni et al., 2022).

While significant efforts have been devoted to understanding the expressive power of graph learning models, the generalization capabilities of such methods are largely an open question. Recent works attempt to study generalization of graph learning models (particularly GNNs) from various perspectives such as extrapolation in feature space (Xu et al., 2021), subgroup fairness (Ma et al., 2021), invariance principle (Wu et al., 2022a), feature propagation (Yang et al., 2023), causality (Wu et al., 2024a), and training dynamics (Yang et al., 2024). However, most of these works focus on the distribution shifts of features and labels. In many critical real-world settings, the training and testing graph topologies can be generated from different distributions (e.g., molecular structures with diverse drug likeness) (Koh et al., 2021; Hu et al., 2021; Bazhenov et al., 2023; Zhang et al., 2023), a phenomenon we refer to as *"topological distribution shift"*. This can be a predominant nature of non-Euclidean data in contrast with commonly studied feature and label shifts in Euclidean space. Despite its practical significance, how to enable graph learning models to generalize under topological shifts still remains unclear.

In this paper, we aim to address generalization under topological shifts through the lens of a physics-inspired graph learning model dubbed as Advective Diffusion Transformer (ADVDIFFORMER). The model is derived from advective diffusion equations which describe a class of continuous message passing process with observed and latent topological structures. On top of this, we connect advective diffusion with a graph Transformer architecture for gener-

---

[1]Eric and Wendy Schmidt Center, Broad Institute of MIT and Harvard [2]Toyota Technological Institute at Chicago [3]Shanghai Jiao Tong University [4]University of Oxford [5]Aithyra. Correspondence to: Qitian Wu <wuqitian@broadinstitute.org>.

*Proceedings of the 42nd International Conference on Machine Learning*, Vancouver, Canada. PMLR 267, 2025. Copyright 2025 by the author(s).

[1]Codes are available at https://github.com/qitianwu/AdvDIFFormer

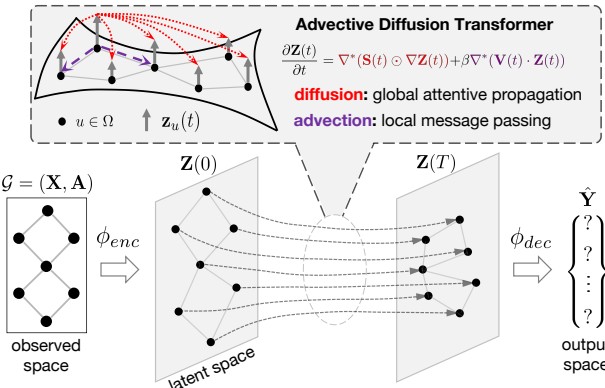

Figure 1: Illustration of Advective Diffusion Transformers.

alization against topological shifts (Fig. 1): the non-local diffusion term (instantiated as global attention) aims to capture latent interactions learned from data; the advection term (instantiated as local message passing) accommodates the topological features pertaining to observed graphs.

To justify the model designs, we show that the closed-form solution of this advective diffusion system possesses the capability to control the generalization error caused by topological shifts, which further guarantees the desired level of generalization. In contrast, commonly used local diffusion models that can be considered as a simplified variant of our model leads to the generalization error whose upper bound exponentially grows with topological shifts.

For implementation, we resort to numerical scheme based on the Padé-Chebyshev theory (Golub & Van Loan, 1989) for efficiently computing the diffusion equation's closed-form solution. Experiments show that our model offers superior generalization performance across various downstream predictive tasks in diverse domains, including information networks, molecular screening, and protein interactions.

## 2. Background and Preliminaries

We recapitulate diffusion equations on manifolds (Freidlin & Wentzell, 1993; Medvedev, 2014) and their connection with graph learning.

**Diffusion on Riemannian manifolds.** Let $\Omega$ denote an abstract domain, which we assume here to be a Riemannian manifold (Eells & Sampson, 1964). A key feature distinguishing an $n$-dimensional Riemannian manifold from a Euclidean space is the fact that it is only *locally* Euclidean, in the sense that at every point $u \in \Omega$ one can construct $n$-dimensional Euclidean *tangent space* $T_u\Omega \cong \mathbb{R}^n$ that locally models the structure of $\Omega$. The collection of such spaces (referred to as the *tangent bundle* and denoted by $T\Omega$) is further equipped with a smoothly-varying inner product

(*Riemannian metric*).

Now consider some quantity (e.g., temperature) as a function of the form $q : \Omega \to \mathbb{R}$, which we refer to as a *scalar field*. Similarly, we can define a *(tangent) vector field* $Q : \Omega \to T\Omega$, associating to every point $u$ on a manifold a tangent vector $Q(u) \in T_u\Omega$, which can be thought of as a local infinitesimal displacement. We use $\mathcal{Q}(\Omega)$ and $\mathcal{Q}(T\Omega)$ to denote the functional spaces of scalar and vector fields, respectively. The *gradient* operator $\nabla : \mathcal{Q}(\Omega) \to \mathcal{Q}(T\Omega)$ takes scalar fields into vector fields representing the local direction of the steepest change of the field. The *divergence* operator is the adjoint of the gradient and maps in the opposite direction, $\nabla^* : \mathcal{Q}(T\Omega) \to \mathcal{Q}(\Omega)$.

A manifold diffusion process models the evolution of a quantity (e.g., chemical concentration) due to its difference across spatial locations on $\Omega$. Denoting by $q(u, t) : \Omega \times [0, \infty) \to \mathbb{R}$ the quantity over time $t$, the process is described by a PDE (*diffusion equation*) (Romeny, 2013):

$$\frac{\partial q(u,t)}{\partial t} = \nabla^* \left( S(u,t) \odot \nabla q(u,t) \right), \ \ t \geq 0, u \in \Omega,$$

with initial conditions $q(u, 0) = q_0(u)$ and possibly additional boundary conditions if $\Omega$ has a boundary. $S$ denotes the *diffusivity* of the domain. It is typical to distinguish between an *isotropic* (location-independent diffusivity), *non-homogeneous* (location-dependent diffusivity $S = s(u) \in \mathbb{R}$), and *anisotropic* (location- and direction-dependent $S(u) \in \mathbb{R}^{n \times n}$) settings. In the cases studied below, we assume the dependence of diffusivity on locations is via a function of the quantity itself, i.e., $S = S(q(u, t))$.

**Diffusion on Graphs.** Recent works adopt diffusion equations as a foundation principle for graph representation learning (Chamberlain et al., 2021a;b; Thorpe et al., 2022; Bodnar et al., 2022; Choi et al., 2023; Rusch et al., 2023; Wu et al., 2024b), employing analogies between calculus on manifolds and graphs. Let $\mathcal{G} = (\mathcal{V}, \mathcal{E})$ be a graph with nodes $\mathcal{V}$ and edges $\mathcal{E}$, represented by the $|\mathcal{V}| \times |\mathcal{V}|$ *adjacency matrix* $\mathbf{A}$. Let $\mathbf{X} = [\mathbf{x}_u]_{u \in \mathcal{V}}$ denote a $|\mathcal{V}| \times D$ matrix of node features, analogous to scalar fields on manifolds. The graph gradient $(\nabla \mathbf{X})_{uv} = \mathbf{x}_v - \mathbf{x}_u$ defines edge features for $(u, v) \in \mathcal{E}$, analogous to vector fields on manifolds. Similarly, the graph divergence of edge features $\mathbf{E} = [\mathbf{e}_{uv}]_{(u,v) \in \mathcal{E}}$, defined as the adjoint $(\nabla^* \mathbf{E})_u = \sum_{v:(u,v) \in \mathcal{E}} \mathbf{e}_{uv}$, produces node features. Diffusion models replace discrete GNN layers with continuous time-evolving node embeddings $\mathbf{Z}(t) = [\mathbf{z}_u(t)]$, where $\mathbf{z}_u(t) : [0, \infty) \to \mathbb{R}^d$ driven by the diffusion equation:

$$\frac{\partial \mathbf{Z}(t)}{\partial t} = \nabla^* \left( \mathbf{S}(\mathbf{Z}(t); \mathbf{A}) \odot \nabla \mathbf{Z}(t) \right), \ \ t \geq 0, \quad (1)$$

with initial conditions $\mathbf{Z}(0) = \phi_{enc}(\mathbf{X})$ where $\phi_{enc}$ is a node-wise MLP encoder and w.l.o.g., the diffusivity

$\mathbf{S}(\mathbf{Z}(t); \mathbf{A})$ over the graph can be defined as a $|\mathcal{V}| \times |\mathcal{V}|$ matrix-valued function dependent on $\mathbf{A}$, which measures the rate of information flows between node pairs. With the graph gradient and divergence, Eqn. 1 becomes

$$\frac{\partial \mathbf{Z}(t)}{\partial t} = (\mathbf{C}(\mathbf{Z}(t); \mathbf{A}) - \mathbf{I})\mathbf{Z}(t), \;\; 0 \leq t \leq T, \quad (2)$$

with initial conditions $\mathbf{Z}(0) = \phi_{enc}(\mathbf{X})$ where $\mathbf{C}(\mathbf{Z}(t); \mathbf{A})$ is a $|\mathcal{V}| \times |\mathcal{V}|$ coupling matrix associated with the diffusivity. Eqn. 2 yields a dynamics from $t = 0$ to an arbitrary given stopping time $T$, where the latter yields node representations for prediction, e.g., $\hat{\mathbf{Y}} = \phi_{dec}(\mathbf{Z}(T))$. The coupling matrix determines the interactions between different nodes in the graph, and its common instantiations include normalized graph adjacency (non-parametric) and learnable attention matrix (parametric), in which cases the finite-difference numerical iterations for solving Eqn. 2 correspond to the discrete propagation layers of common GNNs (Chamberlain et al., 2021a) and graph Transformers (Wu et al., 2023; 2024c) (see Appendix A for details).

It is typical to tacitly make a *closed-world* assumption, i.e., the graph topologies of training and testing data are generated from the same distribution. However, the challenge of generalization arises when the testing topology is different from the training one. In such an *open-world* regime, it still remains unclear how graph diffusion equations and, more broadly, learning-based models on graphs (e.g., GNNs) extrapolate and generalize to new unseen structures.

## 3. Generalization by Advective Diffusion

As a prerequisite for analyzing the generalization behaviors of learning-based models on graphs, we need to characterize how topological shifts occur in nature. In general sense, extrapolation is impossible without any exposure to the new data or prior knowledge about the data-generating mechanism. In our work, we assume testing data is strictly unknown during training, in which case structural assumptions become necessary for authorizing generalization.

### 3.1. Problem Formulation: Data Generation Hypothesis

We present the causal mechanism of graph data generation in Fig. 2 as a hypothesis, inspired by the graph limits (Lovász & Szegedy, 2006; Medvedev, 2014) and random graph models (Snijders & Nowicki, 1997). In graph theory, the topology of a graph $\mathcal{G} = (\mathcal{V}, \mathcal{E})$ can be assumed to be generated by a *graphon* (or continuous graph limit), a random symmetric measurable function $W : [0, 1]^2 \to [0, 1]$, which is an unobserved latent variable. In our work, we generalize this data-generating mechanism to include alongside graph adjacency also node features and labels:

**i)** Each node $u \in \mathcal{V}$ has a latent i.i.d. variable $U_u \sim U[0, 1]$. The *node features* are a random variable $X = [X_u]$ gen-

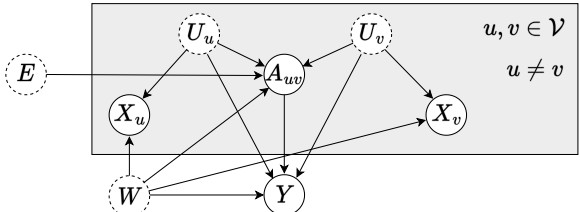

Figure 2: The data-generating mechanism with topological shifts caused by environment $E$. The solid (resp. dashed) nodes represents observed (resp. latent) random variables.

erated from each $U_u$ through a certain node-wise function $X_u = g(U_u; W)$. We denote by matrix $\mathbf{X}$ a particular realization of the random variable $X$.

**ii)** Similarly, the *graph adjacency* $A = [A_{uv}]$ is a random variable generated through a pairwise function $A_{uv} = h(U_u, U_v; W, E)$ additionally dependent on the *environment* $E$. The change of $E$ happens when it transfers from training to testing, resulting in a different distribution of $A$. We denote by $\mathbf{A}$ a particular realization of the adjacency matrix.

**iii)** The *label* $Y$ can be specified in certain forms. As we assume in below, $Y$ is generated by a function over sets, $Y = r(\{U_{v \in \mathcal{V}}\}, A; W)$. Denote by $\mathbf{Y}$ a realization of $Y$.

The above process formalizes the data-generating mechanism behind various data of inter-dependent nature, where the graph data $(\mathbf{X}, \mathbf{A}, \mathbf{Y})$ is generated from the joint distribution $p(X, A, Y|E)$ with a specific environment. The learning problem boils down to finding parameters $\theta$ of a parametric function $\Gamma_\theta(\mathbf{A}, \mathbf{X})$ that establishes the predictive mapping from observed node features $\mathbf{X}$ and graph adjacency $\mathbf{A}$ to the label $\mathbf{Y}$. $\Gamma_\theta$ is typically implemented as a GNN, which is expected to possess sufficient *expressive power* (in the sense that $\exists \theta$ such that $\Gamma_\theta(\mathbf{A}, \mathbf{X}) \approx \mathbf{Y}$) as well as *generalization capability* under topological shifts (i.e., when the observed graph topology varies from training to testing, which in our model amounts to the change in $E$). While significant attention in the literature has been devoted to the former property (Morris et al., 2019; Xu et al., 2019; Bouritsas et al., 2023; Papp et al., 2021; Balcilar et al., 2021; Bodnar et al., 2022); the latter is largely an open question.

### 3.2. Proposed Model: Advective Diffusion Transformers

To deal with the problem formulated in the previous subsection, we next present a new diffusion equation model that offers a provable level of generalization in the data-generating mechanism as stipulated in Sec. 3.1. The model is inspired by a particular class of diffusion equations, called *advective diffusion*, that describe common physical phenomenons driven by both diffusion and advection effects.

**Advective Diffusion Equations.** The classic advective diffusion is commonly used for characterizing physical systems with convoluted quantity transfers, where the term *advection* refers to the evolution caused by the movement of the diffused quantity (Chandrasekhar, 1943). Consider the abstract domain $\Omega$ of our interest defined in Sec. 2, and assume $V(u, t) \in T_u\Omega$ (a vector field in $\Omega$) to denote the velocity of the particle at location $u$ and time $t$. The advective diffusion of the physical quantity $q$ on $\Omega$ is governed by the PDE as (Leveque, 1992): $\frac{\partial q(u,t)}{\partial t} =$

$$\nabla^* \left( S(u, t) \odot \nabla q(u, t) \right) + \beta \nabla^* \left( V(u, t) \cdot q(u, t) \right), \quad (3)$$

where $t \geq 0$, $u \in \Omega$ and $\beta \geq 0$ is a weight for the advection term. For example, if we consider $q(u, t)$ as the water salinity in a river, then Eqn. 3 describes the temporal evolution of salinity at each location that equals to the spatial transfers of both diffusion process (caused by the concentration difference of salt and $S$ reflects the molecular diffusivity in the water) and advection process (caused by the movement of the water and $V$ characterizes the flowing directions).

**Advective Diffusion on Graphs.** Similarly, on a graph $\mathcal{G} = (\mathcal{V}, \mathcal{E})$, we can define the velocity for each node $u$ as a $|\mathcal{V}|$-dimensional vector-valued function $\mathbf{V}(t) = [\mathbf{v}_u(t)]$. We thus have $(\nabla^*(\mathbf{V}(t) \cdot \mathbf{Z}(t)))_u = \sum_{v \in \mathcal{V}} v_{uv}(t)\mathbf{z}_v(t)$ and the advective diffusion equation on graphs:

$$\frac{\partial \mathbf{Z}(t)}{\partial t} = [\mathbf{C}(\mathbf{Z}(t)) + \beta \mathbf{V}(t) - \mathbf{I}] \mathbf{Z}(t), \quad 0 \leq t \leq T. \quad (4)$$

We next instantiate the coupling matrix and the velocity to endow the model with generalizability under topological shifts, by drawing inspirations from physical phenomenons.

○ *Non-local diffusion as global attention.* The diffusion process led by the concentration gradient acts as an internal driving force, where the diffusivity keeps invariant across environments (e.g., the molecular diffusivity stays constant in different rivers). With this intuition in mind, we consider the non-local diffusion operator allowing instantaneous information flows among arbitrary locations (Chasseigne et al., 2006). In the context of learning on graphs, the non-local diffusion can be seen as generalizing the feature propagation to a *complete* or fully-connected (latent) graph (Wu et al., 2023; 2024c), in contrast with common GNNs that allow message passing only between neighboring nodes. Formally speaking, we can define the gradient and divergence operators on a complete graph: $(\nabla \mathbf{X})_{uv} = \mathbf{x}_v - \mathbf{x}_u$ $(u, v \in \mathcal{V})$ and $(\nabla^* \mathbf{E})_u = \sum_{v \in \mathcal{V}} \mathbf{e}_{uv}$ $(u \in \mathcal{V})$. This resonates with the latent interactions among nodes, determined by the underlying data manifold, that induce all-pair information flows over a complete graph and stay invariant w.r.t. the change of $E$. We thus instantiate $\mathbf{C}$ as a global attention matrix that computes the similarities between arbitrary node pairs:

$$\mathbf{C} = [c_{uv}]_{u,v \in \mathcal{V}}, \quad c_{uv} = \frac{\eta(\mathbf{z}_u(0), \mathbf{z}_v(0))}{\sum_{w \in \mathcal{V}} \eta(\mathbf{z}_u(0), \mathbf{z}_w(0))}, \quad (5)$$

where $\eta$ is a learnable pairwise similarity function.

○ *Advection as local message passing.* The advection process driven by the directional movement belongs to an external force, with the velocity depending on contexts (e.g., different rivers). This is analogous to the environment-sensitive graph topology that is informative for prediction in specific environments. We instantiate the velocity as the normalized graph adjacency, i.e., $\mathbf{V} = \mathbf{D}^{-1/2}\mathbf{A}\mathbf{D}^{-1/2}$, reflecting observed structural information, where $\mathbf{D}$ is the diagonal degree matrix of $\mathbf{A}$. Then our advective diffusion model can be formulated as:

$$\frac{\partial \mathbf{Z}(t)}{\partial t} = [\mathbf{C} + \beta \mathbf{V} - \mathbf{I}] \mathbf{Z}(t), \quad 0 \leq t \leq T, \quad (6)$$

with initial conditions $\mathbf{Z}(0) = \phi_{enc}(\mathbf{X})$ where $\beta \in [0, 1]$ is a hyper-parameter. The integration of non-local diffusion (implemented through attention) and advection (implemented as MPNNs) give rise to a new architecture, which we call *Advective Diffusion Transformer* (ADVDIFFORMER).

***Remark.*** Eqn. 6 has a closed-form solution $\mathbf{Z}(t) = e^{-(\mathbf{I}-\mathbf{C}-\beta\mathbf{V})t}\mathbf{Z}(0)$. A special case of $\beta = 0$ (no advection) can be used in situations where the graph structure is not useful. Moreover, one can extend Eqn. 6 to a non-linear equation with time-dependent $\mathbf{C}(\mathbf{Z}(t))$, in which case the equation has no closed-form solution and needs numerical schemes for solving. Similarly to (Di Giovanni et al., 2022), we found in our experiments a simple linear diffusion to be sufficient to yield promising performance. We therefore leave the study of the non-linear variant for the future.

### 3.3. Theoretical Justification

We proceed to analyze the generalization capability of our proposed model w.r.t. topological shifts as defined in Sec 3.1. We are interested in the generalization error of $\Gamma_\theta$ instantiated as the continuous diffusion model in Eqn. 6, when transferring from training data generated with the environment $E_{tr}$ to testing data generated with $E_{te}$. The latter causes varied graph topologies as stipulated in Sec. 3.1.

We denote by $\{(\mathbf{X}^{(i)}, \mathbf{A}^{(i)}, \mathbf{Y}^{(i)})\}_i^{N_{tr}}$ the training data set sized $N_{tr}$ generated from $p(X, A, Y|E = E_{tr})$, and $l(\cdot, \cdot)$ any bounded loss function. The training error (i.e., empirical risk) can be defined as $\mathcal{R}_{emp}(\Gamma_\theta; E_{tr}) \triangleq$

$$\frac{1}{N_{tr}} \sum_{i=1}^{N_{tr}} l(\Gamma_\theta(\mathbf{X}^{(i)}, \mathbf{A}^{(i)}), \mathbf{Y}^{(i)}). \quad (7)$$

Our target is to reduce the generalization error on testing data generated from $p(X, A, Y|E = E_{te})$: $\mathcal{R}(\Gamma_\theta; E_{te}) \triangleq$

$$\mathbb{E}_{(\mathbf{X}', \mathbf{A}', \mathbf{Y}') \sim p(X, A, Y|E=E_{te})}[l(\Gamma_\theta(\mathbf{X}', \mathbf{A}'), \mathbf{Y}')]. \quad (8)$$

Particularly, if $E_{te} = E_{tr}$, the learning setting degrades to the standard one commonly studied in the closed-world

assumption, wherein the in-distribution generalization error has an upper bound (Shalev-Shwartz & Ben-David, 2014):

$$\mathcal{R}(\Gamma_\theta; E_{tr}) - \mathcal{R}_{emp}(\Gamma_\theta; E_{tr}) \leq \mathcal{D}_{in}(\Gamma_\theta, E_{tr}, N_{tr})$$
$$= 2\mathcal{H}(\Gamma_\theta) + O\left(\sqrt{\log(1/\delta)/N_{tr}}\right), \quad (9)$$

where $\mathcal{H}(\Gamma_\theta)$ denotes the Rademacher complexity of the function class induced by $\Gamma_\theta$, and $\mathcal{D}_{in}(\Gamma_\theta, E_{tr}, N_{tr})$ is determined by training data size and model complexity.

When $E_{te} \neq E_{tr}$ that occurs in the open-world regime, i.e., our focused learning setting, the analysis becomes more difficult due to the topological shifts. In the diffusion equation (either Eqn. 6 or 2), the change of graph topologies leads to the change of node representations (solution of the diffusion equation $\mathbf{Z}(T)$). Thereby, the output of the diffusion process can be expressed as $\mathbf{Z}(T; \mathbf{A}) = f(\mathbf{Z}(0), \mathbf{A})$. Our first result below decouples the out-of-distribution generalization gap $\mathcal{R}(\Gamma_\theta; E_{te}) - \mathcal{R}_{emp}(\Gamma_\theta; E_{tr})$ into three error terms.

**Theorem 3.1.** *Assume $l$ and $\phi_{dec}$ are Lipschitz continuous. For any graph data generated with the mechanism of Sec. 3.1, it holds with the probability $1 - \delta$ that the generalization gap of $\Gamma_\theta$ satisfies*

$$|\mathcal{R}(\Gamma_\theta; E_{te}) - \mathcal{R}_{emp}(\Gamma_\theta; E_{tr})| \leq \mathcal{D}_{in}(\Gamma_\theta, E_{tr}, N_{tr})$$
$$+ \underbrace{O(\mathbb{E}_{\mathbf{A}\sim p(A|E_{tr}),\mathbf{A}'\sim p(A|E_{te})}[\|\mathbf{Z}(T; \mathbf{A}') - \mathbf{Z}(T; \mathbf{A})\|_2])}_{\mathcal{D}_{ood-model}(\Gamma_\theta, E_{tr}, E_{te})}$$
$$+ \underbrace{O(\mathbb{E}_{(\mathbf{A},\mathbf{Y})\sim p(A,Y|E_{tr}),(\mathbf{A}',\mathbf{Y}')\sim p(A,Y|E_{te})}[\|\mathbf{Y}' - \mathbf{Y}\|_2])}_{\mathcal{D}_{ood-label}(E_{tr}, E_{te})}.$$

***Remark***. Since $\mathcal{D}_{in}$ is independent of the testing data generated with $E_{te} \neq E_{tr}$, the impact of topological shifts on the out-of-distribution generalization error is largely dependent on $D_{ood-model}$ and $D_{ood-label}$: the former reflects the variation magnitude of $\mathbf{Z}(T; \mathbf{A})$ yielded by $\Gamma_\theta$ w.r.t. varying topologies; the latter measures the difference of labels generated with different environments. Notice that $D_{ood-label}$ is fully determined by the data-generating mechanism, while $D_{ood-model}$ is mainly dependent on the model $\Gamma_\theta$, particularly the sensitivity of node representations w.r.t. topological shifts. We thus next zoom in on the specific design of $\Gamma_\theta$ as Eqn. 6, and the next result shows the upper bound of the change rate of $\mathbf{Z}(T; \mathbf{A})$ w.r.t. variation of graph topologies.

**Theorem 3.2.** *For any graph data generated with the mechanism of Sec. 3.1, if $g$ is injective, then the model Eqn. 6 can reduce the variation magnitude of the node representation $\|\mathbf{Z}(T; \mathbf{A}') - \mathbf{Z}(T; \mathbf{A})\|_2$ to any order $O(\psi(\|\Delta\tilde{\mathbf{A}}\|_2))$ where $\psi$ denotes an arbitrary polynomial function, $\Delta\tilde{\mathbf{A}} = \tilde{\mathbf{A}}' - \tilde{\mathbf{A}}$ and $\tilde{\mathbf{A}} = \mathbf{D}^{-1/2}\mathbf{A}\mathbf{D}^{-1/2}$.*

This suggests that the advective diffusion model is capable of controlling the change rate of node representations to arbitrary rates w.r.t. $\|\Delta\tilde{\mathbf{A}}\|_2$. The injectivity of $g$ is a mild

condition since $g$ establishes a mapping from a smooth and compact latent space to a high-dimensional space. Applying Theorem 3.1 we have the generalization error of the advective diffusion model.

**Corollary 3.3.** *On the same condition of Theorem 3.1 and 3.2, the model-dependent generalization error bound of Eqn 6 can be reduced to arbitrary polynomial orders w.r.t. topological shifts, i.e., $\mathcal{D}_{ood-model}(\Gamma_\theta, E_{tr}, E_{tr}) = O(\mathbb{E}_{\mathbf{A}\sim p(A|E_{tr}),\mathbf{A}'\sim p(A|E_{te})}[\psi(\|\Delta\tilde{\mathbf{A}}\|_2)])$.*

This implies that the generalization error of the model in Eqn. 6 can be controlled within an arbitrary rate w.r.t. $\|\Delta\tilde{\mathbf{A}}\|_2$. The model has provable capacity for achieving a desired level of generalization with topological shifts. To verify the efficacy of the model, we consider synthetic data that simulates the topological shifts as defined in Sec. 3.1 and investigate into three types of topological shifts as shown in Fig. 3 (see experimental details in Sec. 5.1). We found that our model (ADVDIFFORMER-I and ADVDIFFORMER-S whose implementation is presented in Sec. 4) keeps the testing error nearly constant as $\|\Delta\tilde{\mathbf{A}}\|_2$ increases.

### 3.4. Comparisons with Other Models

To further illuminate the effectiveness of the proposed model, we next compare with two related models that are commonly adopted and can be considered as the simplified variants of our model.

**Local Diffusion Model.** We first consider a typical model instantiation, i.e., local diffusion equation on graphs, wherein the model discards the advection term in Eqn. 6 and degrades to Eqn. 2 with the coupling matrix dependent on $\mathbf{A}$. In such a situation, the propagation of node signals is constrained within connected neighbored nodes. The common choice for the coupling matrix is the symmetric normalized graph adjacency matrix $\tilde{\mathbf{A}} = \mathbf{D}^{-1/2}\mathbf{A}\mathbf{D}^{-1/2}$. In this case, the finite-difference iteration for solving Eqn. 2 would induce the discrete propagation layers akin to the message passing rule of SGC (Wu et al., 2019) and GCN (Kipf & Welling, 2017) if the feature transformation and nonlinearity are neglected (see more illustration in Appendix A). Given the constant coupling matrix $\mathbf{C}$, Eqn. 2 has a closed-form solution $\mathbf{Z}(t) = e^{-(\mathbf{I}-\mathbf{C})t}\mathbf{Z}(0)$. However, unlike the advective diffusion model, the change rate of $\mathbf{Z}(T; \mathbf{A})$ w.r.t. $\Delta\tilde{\mathbf{A}} = \tilde{\mathbf{A}}' - \tilde{\mathbf{A}}$ produced by the local diffusion model has an exponential upper bound.

**Proposition 3.4.** *For local diffusion model (defined by Eqn. 2) with the coupling matrix $\mathbf{C} = \mathbf{D}^{-1/2}\mathbf{A}\mathbf{D}^{-1/2}$ or $\mathbf{C} = \mathbf{D}^{-1}\mathbf{A}$, the yielded node representation satisfies $\|\mathbf{Z}(T; \mathbf{A}') - \mathbf{Z}(T; \mathbf{A})\|_2 = O(\|\Delta\tilde{\mathbf{A}}\|_2 \exp(\|\Delta\tilde{\mathbf{A}}\|_2 T))$.*

The label prediction $\hat{\mathbf{Y}} = \phi_{dec}(\mathbf{Z}(T; \mathbf{A}))$ could be highly sensitive to the change of the graph topology. Pushing fur-

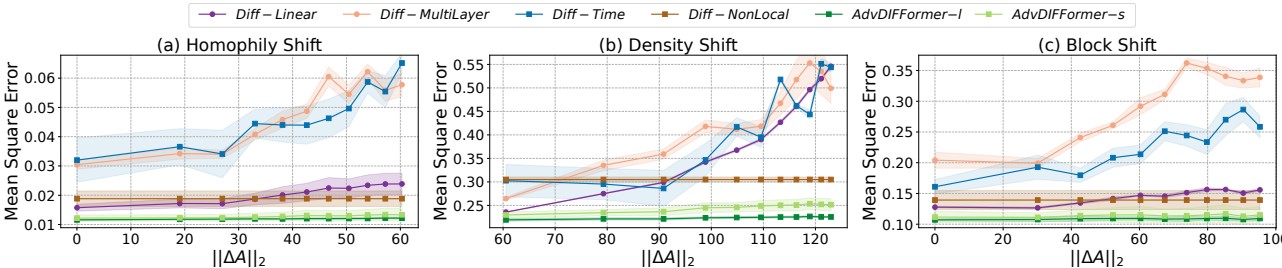

Figure 3: Testing errors (y-axix) w.r.t. differences in graph topologies (x-axis) on synthetic datasets that simulate the topological distribution shifts according to the data generation hypothesis of Fig. 2.

ther, we have the following corollary on the generalization capability of local diffusion models under topological shifts.

**Corollary 3.5.** *Under the same condition as in Theorem 3.1, for diffusion models Eqn. 2 with the normalized graph adjacency as the coupling matrix, the model-dependent generalization error on testing data generated with $E_{te} \neq E_{tr}$ has an upper bound: $\mathcal{D}_{ood-model}(\Gamma_\theta, E_{tr}, E_{tr}) = O(\mathbb{E}_{\mathbf{A}\sim p(A|E_{tr}), \mathbf{A}'\sim p(A|E_{te})}[\|\Delta\tilde{\mathbf{A}}\|_2 \exp(\|\Delta\tilde{\mathbf{A}}\|_2 T)]).$*

By definition in Sec. 3.1, the graph adjacency is a realization of a random variable $A = h(U_u, U_v; W, E)$ dependent on a varying environment $E$. Topological shifts caused by different distributions of $\mathbf{A}$'s between training and testing environments may result in large $\mathcal{D}_{ood-model}$. [2] This result together with Theorem 3.1 suggests that local diffusion models may lead to undesirably poor generalization in cases where models are expected to be insensitive to the change of topologies. For example, for situations where the ground-truth labels do not dramatically change with topological shifts (i.e., $\mathcal{D}_{ood-label}$ is small), local diffusion models may induce large $\mathcal{D}_{ood-model}$ that prejudices generalization. The above conclusion can be extended to models with layer-wise feature transformations and non-linearity (see Appendix B.6 for illustration). While the upper bound result does not mean the exponentially growing error would definitely happen in practice, our empirical comparison in Fig. 3 shows that the generalization error of local diffusion models (*Diff-Linear*, *Diff-MultiLayer* and *Diff-Time*) on test data grows super-linearly w.r.t. $\|\Delta\tilde{\mathbf{A}}\|_2$ across three types of topological shifts (see experimental details in Sec. 5.1).

**Non-Local Diffusion Model.** We next discuss the non-local diffusion model without the advection term, which as previously mentioned can be essentially treated as a generalization of local diffusion to a complete or fully-connected graph. The corresponding diffusion equation still exhibits the form of Eqn. 2 but allows non-zero entries for arbitrary $(u, v)$'s in the coupling matrix to accommodate the all-pair information flows. Then we can easily derive a result that if $Y$ is conditionally independent from $A$ with given $\{U_u\}_{u\in\mathcal{V}}$ in the data generation hypothesis of Sec. 3.1, the non-local diffusion model (i.e., Eqn. 2 with the attention-based coupling matrix) leads to the generalization gap

$$\mathcal{R}(\Gamma_\theta; E_{te}) - \mathcal{R}_{emp}(\Gamma_\theta; E_{tr}) \leq \mathcal{D}_{in}(\Gamma_\theta, E_{tr}, N_{tr}), \quad (10)$$

which holds with the probability $1 - \delta$. The assumption of conditional independence between $Y$ and $A$, however, can be violated in many situations where labels strongly correlate with observed graph structures. Furthermore, the performance on testing data (i.e., what we care about) depends on both the model's expressiveness and generalization. The non-local diffusion alone, discarding any observed topology, has insufficient expressiveness for capturing the structural information. By contrast, the advective diffusion model proposed in Sec. 3.2 that accommodates the observed structures can provably generalize under topological shifts without the conditional independence assumption between $Y$ and $A$. This makes the advective diffusion model more powerful in real cases. Furthermore, as empirically validated in Fig. 3, the non-local diffusion model (*Diff-NonLocal*) indeed yields comparably stable yet obviously inferior performance to our models (ADVDIFFORMER-I and ADVDIFFORMER-S).

## 4. Model Implementation

The remaining question concerning model implementation boils down to how to solve the advective diffusion equation Eqn. 6. One straightforward solution is to harness the scheme adopted by (Chen et al., 2018) for back-propagation through PDE dynamics. However, since it is known that the equation has a closed-form solution $e^{-(\mathbf{I}-\mathbf{C}-\beta\mathbf{V})t}$, we resort to a implementation-wise simpler method by computing the solution instead of numerically solving the equation. Nevertheless, direct computation of the matrix exponential through eigendecomposition is computationally intractable for large matrices. As an alternative, we leverage numer-

---

[2] The influence of topology variation is inherently associated with $h$. For example, if one considers $h$ as the stochastic block model (Snijders & Nowicki, 1997), then the change of $E$ may lead to generated graph data with different edge probabilities. In the case of real-world data with intricate topological patterns, the functional forms of $h$ can be more complex, consequently inducing different types of topological shifts.

ical techniques based on series expansion that produces two model versions. Due to space limit, we present the main ideas in this subsection and defer details on model implementation to Appendix D.1.

**ADVDIFFORMER-I** uses a numerical method based on the extension of Padé-Chebyshev theory to rational fractions (Golub & Van Loan, 1989; Gallopoulos & Saad, 1992), which has shown empirical success in 3D shape analysis (Patané, 2014). The matrix exponential is approximated by solving multiple linear systems (see more details and derivations in Appendix C) and we generalize it as a multi-head network where each head propagates in parallel:

$$\mathbf{L}_h = (1 + \theta)\mathbf{I} - \mathbf{C}_h - \beta\mathbf{V}, \ \ h = 1, \cdots, H,$$
$$\mathbf{Z}(T) \approx \sum_{h=1}^{H} \phi_{FC}^{(h)}(\texttt{linsolver}(\mathbf{L}_h, \mathbf{Z}(0))). \quad (11)$$

The `linsolver` computes the matrix inverse $\mathbf{Z}_h = (\mathbf{L}_h)^{-1}\mathbf{Z}(0)$ and can be efficiently implemented via `torch.linalg.solve()` that enables automated differentiation. Each head contains propagation with the pre-computed attention $\mathbf{C}_h$ and node-wise transformation $\phi_{FC}^{(h)}$.

**ADVDIFFORMER-S** resorts to approximation by finite geometric series (see Appendix C for derivations):

$$\mathbf{P}_h = \mathbf{C}_h + \beta\tilde{\mathbf{A}}, \ \ h = 1, \cdots, H,$$
$$\mathbf{Z}(T) \approx \sum_{h=1}^{H} \phi_{FC}^{(h)}([\mathbf{Z}(0), \mathbf{P}_h\mathbf{Z}(0), \cdots, (\mathbf{P}_h)^K\mathbf{Z}(0)]). \quad (12)$$

This model aggregates $K$-order propagated results with the propagation matrix $\mathbf{P}_h$ in each head. One advantage of this model version lies in its good scalability with linear complexity w.r.t. the number of nodes in the feed-forward computation (see detailed illustration in Appendix D.1.2).

# 5. Experiments

We apply our model to a wide variety of downstream tasks of disparate scales and granularities that involve topological shifts led by distinct factors. Due to the diversity of datasets and tasks, the competing models that are applicable to specific cases can vary case by case, so the goal of our experiments is to showcase the wide applicability and superiority of ADVDIFFORMER against commonly used GNNs and graph Transformers as well as bespoke methods tailored for specific tasks. In the following, we delve into each case separately with the overview of experimental setup and discussions. More detailed dataset information is provided in Appendix E.1. Details on baselines and hyper-parameters are deferred to Appendix E.2 and E.3, respectively.

## 5.1. Synthetic Datasets

To validate our model and theoretical analysis, we create synthetic datasets simulating the data generation hypothesis

Table 1: Results on `Arxiv` and `Twitch`, where we use time and spatial contexts for data splits, respectively. We report the Accuracy (↑) for three testing sets of `Arxiv` and average ROC-AUC (↑) for all testing graphs of `Twitch` (results for each case are reported in Appendix F.1). Top performing methods are marked as first/second/third. OOM indicates out-of-memory error.

| | Arxiv (2018) | Arxiv (2019) | Arxiv (2020) | Twitch (avg) |
|---|---|---|---|---|
| MLP | 49.91 ± 0.59 | 47.30 ± 0.63 | 46.78 ± 0.98 | 61.12 ± 0.16 |
| GCN | 50.14 ± 0.46 | 48.06 ± 1.13 | 46.46 ± 0.85 | 59.76 ± 0.34 |
| GAT | 51.60 ± 0.43 | 48.60 ± 0.28 | 46.50 ± 0.21 | 59.14 ± 0.72 |
| SGC | 51.40 ± 0.10 | 49.15 ± 0.16 | 46.94 ± 0.29 | 60.86 ± 0.13 |
| GDC | 51.53 ± 0.42 | 49.02 ± 0.51 | 47.33 ± 0.60 | 61.36 ± 0.10 |
| GRAND | 52.45 ± 0.27 | 50.18 ± 0.18 | 48.01 ± 0.24 | 61.65 ± 0.23 |
| A-DGNs | 50.91 ± 0.41 | 47.54 ± 0.61 | 45.79 ± 0.39 | 60.11 ± 0.09 |
| CDE | 50.54 ± 0.21 | 47.31 ± 0.52 | 45.32 ± 0.26 | 60.69 ± 0.10 |
| GraphTrans | OOM | OOM | OOM | 61.65 ± 0.23 |
| GraphGPS | 51.11 ± 0.19 | 48.91 ± 0.34 | 46.46 ± 0.95 | 62.13 ± 0.34 |
| DIFFormer | 50.45 ± 0.94 | 47.37 ± 1.58 | 44.30 ± 2.02 | 62.11 ± 0.11 |
| **ADVDIFFORMER-S** | 53.41 ± 0.48 | 51.53 ± 0.60 | 49.64 ± 0.54 | 62.51 ± 0.07 |

in Sec. 3.1. We instantiate $h$ as a stochastic block model which generates edges $A_{uv}$ according to block numbers ($b$), intra-block edge probability ($p_1$) and inter-block edge probability ($p_2$). Then we study three types of topological distribution shifts: **homophily shift** (changing $p_2$ with fixed $p_1$); **density shift** (changing $p_1$ and $p_2$); and **block shift** (varying $b$). The predictive task is node regression. More details on data generation are presented in Appendix E.1.1.

Fig. 3 plots the testing error (i.e., Mean Square Error) w.r.t. differences in graph topologies $\|\Delta\mathbf{A}\|_2$ (i.e., the gap between training and testing graphs) in three cases. We compare our model (ADVDIFFORMER-I and ADVDIFFORMER-S) with other diffusion-based models as competitors. The latter includes *Diff-Linear* (graph diffusion with constant $\mathbf{C}$), *Diff-MultiLayer* (the extension of *Diff-Linear* with intermediate feature transformations), *Diff-Time* (graph diffusion with time-dependent $\mathbf{C}(\mathbf{Z}(t))$) and *Diff-NonLocal* (non-local diffusion with the global attention-based $\mathbf{C}(\mathbf{Z}(t))$). The results show that three local graph diffusion models exhibit clear performance degradation, i.e., the regression error grows super-linearly w.r.t. topological shifts, while our two models yield consistently low error across environments. In contrast, the non-local diffusion model produces comparably stable performance yet inferior to our models due to its ignorance of the useful information in input graphs. These empirical observations are consistent with our theoretical results presented in Sec. 3.3 and 3.4.

## 5.2. Real-World Datasets

We next evaluate ADVDIFFORMER on real-world datasets with more complex topological shifts concerning non-Euclidean data in a diverse set of applications.

**Information Networks**. We first consider citation networks `Arxiv` (Hu et al., 2020) and social networks

Table 2: Results of RMSE (↓) for node regression and edge regression on dynamic networks of protein-protein interactions.

| | Node Regression | | | Edge Regression | | |
|---|---|---|---|---|---|---|
| | **Valid** | **Test Average** | **Test Worst** | **Valid** | **Test Average** | **Test Worst** |
| **MLP** | $0.768 \pm 0.011$ | $0.672 \pm 0.014$ | $0.768 \pm 0.014$ | $0.150 \pm 0.004$ | $0.192 \pm 0.003$ | $0.204 \pm 0.003$ |
| **GCN** | $1.791 \pm 0.023$ | $1.308 \pm 0.013$ | $1.797 \pm 0.007$ | $0.185 \pm 0.003$ | $0.196 \pm 0.001$ | $0.213 \pm 0.001$ |
| **GAT** | $1.255 \pm 0.022$ | $1.057 \pm 0.030$ | $1.708 \pm 0.067$ | $0.210 \pm 0.010$ | $0.204 \pm 0.006$ | $0.216 \pm 0.010$ |
| **SGC** | $1.622 \pm 0.004$ | $1.154 \pm 0.006$ | $1.616 \pm 0.002$ | $0.193 \pm 0.000$ | $0.191 \pm 0.001$ | $0.209 \pm 0.001$ |
| **GraphTrans** | $3.798 \pm 1.146$ | $3.203 \pm 0.889$ | $3.795 \pm 1.123$ | $0.189 \pm 0.005$ | $0.189 \pm 0.008$ | $0.202 \pm 0.003$ |
| **GraphGPS** | $0.713 \pm 0.050$ | $0.671 \pm 0.040$ | $0.803 \pm 0.081$ | $0.168 \pm 0.004$ | $0.182 \pm 0.007$ | $0.216 \pm 0.019$ |
| **DIFFormer** | $0.672 \pm 0.046$ | $0.637 \pm 0.034$ | $0.710 \pm 0.028$ | $0.171 \pm 0.007$ | $0.183 \pm 0.005$ | $0.197 \pm 0.003$ |
| **ADVDIFFORMER-I** | $0.681 \pm 0.010$ | $0.643 \pm 0.019$ | $0.679 \pm 0.021$ | $0.159 \pm 0.002$ | $0.166 \pm 0.006$ | $0.184 \pm 0.011$ |
| **ADVDIFFORMER-S** | $0.547 \pm 0.040$ | $0.574 \pm 0.028$ | $0.644 \pm 0.040$ | $0.156 \pm 0.006$ | $0.167 \pm 0.004$ | $0.188 \pm 0.010$ |

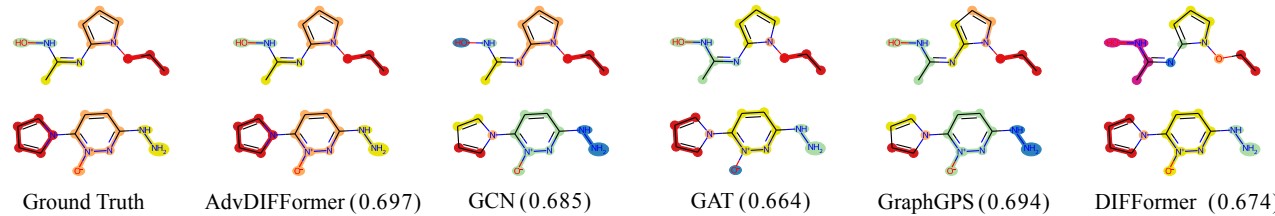

| Ground Truth | AdvDIFFormer (0.697) | GCN (0.685) | GAT (0.664) | GraphGPS (0.694) | DIFFormer (0.674) |
|---|---|---|---|---|---|

Figure 4: Testing cases for molecular mapping operators generated by different models with averaged testing Accuracy (↑) reported. The task is to generate subgraph-level partitions (marked by different colors) resembling the ground-truth.

Twitch (Rozemberczki et al., 2021) with graph sizes ranging from 2K to 0.2M, where we use the scalable version ADVDIFFORMER-S. To introduce topological shifts, we partition the data according to publication years and geographic information for Arxiv and Twitch, respectively. The predictive task is node classification, and we follow the common practice comparing Accuracy (resp. ROC-AUC) for Arxiv (resp. Twitch). We compare with three types of state-of-the-art baselines: (i) **classical GNNs** (*GCN* (Kipf & Welling, 2017), *GAT* (Velickovic et al., 2018) and *SGC* (Wu et al., 2019)); (ii) **diffusion-based GNNs** (*GDC* (Klicpera et al., 2019), *GRAND* (Chamberlain et al., 2021a), *A-DGNs* (Gravina et al., 2023) and *CDE* (Zhao et al., 2023)), and (iii) **graph Transformers** (*GraphTrans* (Wu et al., 2021), *GraphGPS* (Rampásek et al., 2022), and the diffusion-based *DIFFormer* (Wu et al., 2023)). Appendix E.2 presents detailed descriptions for these models. Table 1 reports the results, showing that our model offers significantly superior generalization for node classification.

**Protein Interactions**. We then test on protein-protein interactions (Fu & He, 2022). Each node denotes a protein with a time-aware gene expression value and the edges indicate co-expressed protein pairs at each time. The dataset consists of 12 dynamic networks each of which is obtained by one protein identification method and records the metabolic cycles of yeast cells. The networks have distinct topological features (e.g., distribution of cliques) (Fu & He, 2022), and we use 6/1/5 networks for train/valid/test. To test the gen-

eralization of the model in different scenarios, we consider two predictive tasks: i) node regreesion for gene expression values (measured by RMSE), and 2) edge regression for predicting the co-expression correlation coefficients (measured by RMSE). Table 2 reports the averaged scores and worst-case scores across all testing graphs. The results show that ADVDIFFORMER-S and ADVDIFFORMER-I are ranked in the top three (resp. two) models in terms of the averaged (resp. worst-case) testing performance. Moreover, ADVDIFFORMER-S performs better in node regression tasks, while ADVDIFFORMER-I exhibits (slightly) better competitiveness for edge regression. The possible reason might be that ADVDIFFORMER-I can better exploit high-order structural information as the matrix inverse can be treated as ADVDIFFORMER-S with $K \to \infty$.

**Molecular Mapping Operator Generation**. We next consider the generation of molecular coarse-grained mapping operators, an important step for molecular dynamics simulation, aiming to find a representation of how atoms are grouped in a molecule (Li et al., 2020). The task is a graph segmentation problem which can be modeled as predicting edges that indicate where to partition the graph. We use the relative molecular mass to split the data and test how the model extrapolates to larger molecules. Fig. 4 compares the testing cases (with more cases shown in Appendix F.1) generated by different models, which shows the more accurate estimation of our model (we use ADVDIFFORMER-S for experiments) that demonstrates desired generalization.

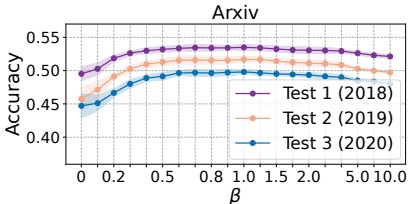 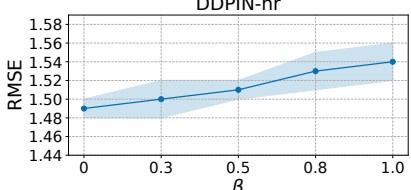 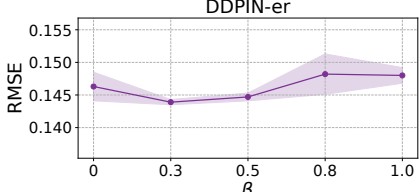

Figure 5: Analysis of $\beta$ on `Arxiv` and node regression (nr) and edge regression (er) tasks on `DPPIN`.

**Hyper-Parameter Analysis.** The hyper-parameter $\beta$ controls the importance weight for the advection term. Fig. 5 shows the model performance of ADVDIFFORMER-S on `Arxiv` and `DPPIN` with different $\beta$'s. We found that the optimal settings for $\beta$ can be different across datasets and tasks. For node classification on `Arxiv`, the model gives the best performance with $\beta \in [0.7, 1.0]$. The performance degrades when $\beta$ is too small ($<0.5$) or too large ($>2.0$). The reason could be that the graph structural information is useful for the predictive task on `Arxiv` yet too much emphasis on the graph structure can lead to undesired generalization. Differently, for `DPPIN`, we found that using smaller $\beta$ can bring up more satisfactory performance across node regression and edge regression tasks. In particular, setting $\beta = 0$, in which case the advection term is completely dropped, can yield optimal performance for the node regression task. This is possibly because the graph structure is uninformative and pure global attention can learn generalizable topological patterns from latent interactions. To sum up, in practice, the model enables much flexibility for adjusting the weight on the advection effect (the importance of observed structural information) to accommodate the diversity of graph-structured data. More hyper-parameter analysis (w.r.t. $\theta$ and $K$) is deferred to Appendix F.2.

**Ablation Studies.** We defer ablation studies on the diffusion and advection terms of our model to Appendix F.2.

## 6. Conclusions

This paper studies generalization with topological shifts, a largely open question in machine learning, and the insights in this work open new possibilities of leveraging diffusion PDEs as principled guidance for navigating generalizable neural network models. As exemplified in this work, our proposed Advective Diffusion Transformer, inspired by advective diffusion equations, has provable potentials for generalization and shows superior performance in various downstream tasks across different scenarios.

## Impact Statement

This paper presents work whose goal is to advance the current understandings for the generalization of neural network models. In general sense, improving the generalizability of the model is important for many aspects associated with AI's societal responsibilities, such as addressing the observational bias in training data and promoting the fairness of the outcome on the test set. Specifically speaking, our paper explores the generalization capability of neural network models operated on varying graph topologies, which has implications on several significant applications such as social network analysis, drug discovery and healthcare. The results and methodology presented in this work can shed lights on enhancing the trustworthiness and reliability of machine learning models in the open world regime.

## Acknowledgement

QW is supported in part by funding from the Eric and Wendy Schmidt Center at the Broad Institute of MIT and Harvard. MB is partially supported by the EPSRC Turing AI World-Leading Research Fellowship No. EP/X040062/1 and EPSRC AI Hub No. EP/Y028872/1.

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

# A. Connection between Diffusion Equations and Message Passing

In this section, we provide a systematically introduction on the fundamental connections between graph diffusion equations and neural message passing, as supplementary technical background for our analysis and methodology presented in the main text. Consider graph diffusion equations of the generic form

$$\frac{\partial \mathbf{Z}(t)}{\partial t} = (\mathbf{C}(\mathbf{Z}(t); \mathbf{A}) - \mathbf{I})\mathbf{Z}(t), \ \ 0 \le t \le T, \ \ \text{with initial conditions } \mathbf{Z}(0) = \phi_{enc}(\mathbf{X}). \tag{13}$$

As demonstrated by existing works, e.g., (Chamberlain et al., 2021a), using finite-difference numerical schemes for solving Eqn. 13 would induce the message passing neural networks of various forms. The latter is recognized as the common paradigm in modern graph neural networks and Transformers whose layer-wise updating aggregates the embeddings of other nodes to compute the embeddings for the next layer.

## A.1. Graph Neural Networks as Local Diffusion

Consider the explicit Euler's scheme as the commonly used finite-difference method for approximately solving the differential equations, and Eqn. 13 will induce the discrete iterations with step size $\tau$:

$$\frac{\mathbf{Z}^{(k+1)} - \mathbf{Z}^{(k)}}{\tau} \approx (\mathbf{C}(\mathbf{Z}^{(k)}; \mathbf{A}) - \mathbf{I})\mathbf{Z}^{(k)}. \tag{14}$$

With some re-arranging we have

$$\mathbf{Z}^{(k+1)} = (1 - \tau)\mathbf{Z}^{(k)} + \tau\mathbf{C}(\mathbf{Z}^{(k)}; \mathbf{A})\mathbf{Z}^{(k)}, \tag{15}$$

with the initial states $\mathbf{Z}^{(0)} = \phi_{enc}(\mathbf{X})$. The above updating equation gives one-layer update through residual connection and propagation with $\mathbf{C}(\mathbf{Z}^{(k)}; \mathbf{A})$. There are some well-known graph neural network architectures that can be derived with different instantiations of the coupling matrix.

**Simplifying Graph Convolution (SGC).** If one considers $\mathbf{C}(\mathbf{Z}^{(k)}; \mathbf{A}) = \tilde{\mathbf{A}} = \mathbf{D}^{-1/2}\mathbf{A}\mathbf{D}^{-1/2}$, then we will get the one-layer updating rule:

$$\mathbf{Z}^{(k+1)} = (1 - \tau)\mathbf{Z}^{(k)} + \tau\mathbf{D}^{-1/2}\mathbf{A}\mathbf{D}^{-1/2}\mathbf{Z}^{(k)}. \tag{16}$$

This can be seen as one-layer propagation of SGC (Wu et al., 2019) with residual connection, and when $\tau = 1$ it becomes exactly the SGC layer. Since SGC model does not involve feature transformation layers and non-linearity throughout the message passing, one often uses a pre-computed propagation matrix for one-step convolution that is much faster than the multi-layer convolution:

$$\mathbf{Z}^{(K)} = \mathbf{P}^K\mathbf{Z}^{(0)}, \ \ \mathbf{P} = (1 - \tau)\mathbf{I} + \tau\mathbf{D}^{-1/2}\mathbf{A}\mathbf{D}^{-1/2}. \tag{17}$$

**Graph Convolution Networks (GCN).** The GCN network inserts feature transformation layers in-between the propagation layers. This can be achieved by considering $K$ stacked piece-wise diffusion equations, where the $k$-th dynamics is given by the differential equation with time boundaries:

$$\frac{\partial \mathbf{Z}(t; k)}{\partial t} = (\mathbf{C} - \mathbf{I})\mathbf{Z}(t; k), \ \ t \in [t_{k-1}, t_k], \ \ \text{with initial conditions } \mathbf{Z}(t_{k-1}; k) = \phi_{int}^{(k)}(\mathbf{Z}(t_{k-1}; k - 1)), \tag{18}$$

where $\phi_{int}^{(k)}$ denotes the node-wise feature transformation of the $k$-th layer. Assume $\mathbf{C} = \mathbf{D}^{-1/2}\mathbf{A}\mathbf{D}^{-1/2}$. Then consider one-step feed-forward of the explicit Euler scheme for Eqn. 18, and one can obtain the updating rule at the $k$-th layer:

$$\mathbf{Z}^{(k+1)} = \phi_{int}^{(k+1)}\left((1 - \tau)\mathbf{Z}^{(k)} + \tau\mathbf{D}^{-1/2}\mathbf{A}\mathbf{D}^{-1/2}\mathbf{Z}^{(k)}\right). \tag{19}$$

This corresponds to one GCN layer (Kipf & Welling, 2017) if one considers $\phi_{int}^{(k+1)}$ as a fully-connected neural layer with ReLU activation and simply sets $\tau = 1$.

**High-Order Propagation.** Besides the explicit numerical scheme, one can also utilize the implicit scheme and multi-step schemes (e.g., Runge-Kutta) for solving the diffusion equation, and the induced updating form will involve high-order information (Chamberlain et al., 2021a).

### A.2. Graph Transformers as Non-Local Diffusion

The original architectures of Transformers (Vaswani et al., 2017) involve self-attention layers as the key module, where the attention measures the pairwise influence between arbitrary token pairs in the input. There are recent works, e.g., (Dwivedi & Bresson, 2020; Ying et al., 2021; Wu et al., 2021; Rampásek et al., 2022; Wu et al., 2022b) transferring the Transformer architectures originally designed for sequence inputs into graph-structured data, and the attention is computed for arbitrary node pairs in the graph, which can be seen as a counterpart of non-local diffusion (Wu et al., 2023; 2024c). In specific, the coupling matrix allows non-zero entries for arbitrary location pairs and can be instantiated as a global attention network. Then using the explicit Euler's scheme as Eqn. 15 we can obtain the self-attention propagation layer of common Transformers:

$$\mathbf{Z}^{(k+1)} = (1 - \tau)\mathbf{Z}^{(k)} + \tau\mathbf{C}^{(k)}\mathbf{Z}^{(k)}, \quad c_{uv}^{(k)} = \frac{\eta(\mathbf{z}_u^{(k)}, \mathbf{z}_v^{(k)})}{\sum_{w \in \mathcal{V}} \eta(\mathbf{z}_u^{(k)}, \mathbf{z}_w^{(k)})}. \tag{20}$$

For obtaining the fully-connected layers and non-linear activations adopted in Transformers, one can inherit the spirit of GCN and extend the diffusion model to $K$ piece-wise equations as Eqn. 18.

## B. Proofs for Technical Results

### B.1. Proof for Theorem 3.1

According to the data generation hypothesis in Fig. 2, for given node latents $U_u$'s, we can decompose the joint distribution into (we omit all conditions on $U$ for brevity)

$$p(X, A, Y|E) = p(X|E)p(A|E)p(Y|A, E). \tag{21}$$

Also, by definition in Sec. 3.1 we have

$$p(X|E = E_{tr}) = p(X|E = E_{te}), \tag{22}$$

$$p(Y|A, E = E_{tr}) = p(Y|A, E = E_{te}). \tag{23}$$

Therefore we have $p(X, A, Y|E) = p(X)p(A|E)p(Y|A)$. We next consider the gap between $\mathcal{R}(\Gamma_\theta; E_{tr})$ and $\mathcal{R}(\Gamma_\theta; E_{te})$:

$$
\begin{aligned}
&\left|\mathcal{R}(\Gamma_\theta; E_{te}) - \mathcal{R}(\Gamma_\theta; E_{tr})\right| \\
=& \left|\mathbb{E}_{(\mathbf{X}', \mathbf{A}', \mathbf{Y}') \sim p(X, A, Y|E=E_{te})}[l(\Gamma_\theta(\mathbf{X}', \mathbf{A}'), \mathbf{Y}')] - \mathbb{E}_{(\mathbf{X}, \mathbf{A}, \mathbf{Y}) \sim p(X, A, Y|E=E_{tr})}[l(\Gamma_\theta(\mathbf{X}, \mathbf{A}), \mathbf{Y})]\right| \\
=& \left|\mathbb{E}_{\mathbf{X}' \sim p(X), \mathbf{A}' \sim p(A|E=E_{te}), \mathbf{Y}' \sim p(Y|A=\mathbf{A}'))}[l(\Gamma_\theta(\mathbf{X}', \mathbf{A}'), \mathbf{Y}')] \right. \\
& \left. - \mathbb{E}_{\mathbf{X} \sim p(X), \mathbf{A} \sim p(A|E=E_{tr}), \mathbf{Y} \sim p(Y|A=\mathbf{A}))}[l(\Gamma_\theta(\mathbf{X}, \mathbf{A}), \mathbf{Y})]\right| \\
\leq& \left|\mathbb{E}_{\mathbf{X}' \sim p(X), \mathbf{A}' \sim p(A|E=E_{te}), \mathbf{Y}' \sim p(Y|A=\mathbf{A}'))}[l(\Gamma_\theta(\mathbf{X}', \mathbf{A}'), \mathbf{Y}')] \right. \\
& \left. - \mathbb{E}_{\mathbf{X} \sim p(X), \mathbf{A} \sim p(A|E=E_{tr}), \mathbf{A}' \sim p(A|E=E_{te}), \mathbf{Y}' \sim p(Y|A=\mathbf{A}'))}[l(\Gamma_\theta(\mathbf{X}, \mathbf{A}), \mathbf{Y}')]\right| \\
& + \left|\mathbb{E}_{\mathbf{X} \sim p(X), \mathbf{A} \sim p(A|E=E_{tr}), \mathbf{A}' \sim p(A|E=E_{te}), \mathbf{Y}' \sim p(Y|A=\mathbf{A}'))}[l(\Gamma_\theta(\mathbf{X}, \mathbf{A}), \mathbf{Y}')] \right. \\
& \left. - \mathbb{E}_{\mathbf{X} \sim p(X), \mathbf{A} \sim p(A|E=E_{tr}), \mathbf{Y} \sim p(Y|A=\mathbf{A}))}[l(\Gamma_\theta(\mathbf{X}, \mathbf{A}), \mathbf{Y})]\right| \\
=& \left|\mathbb{E}_{\mathbf{X} \sim p(X), \mathbf{A} \sim p(A|E=E_{tr}), \mathbf{A}' \sim p(A|E=E_{te}), \mathbf{Y}' \sim p(Y|A=\mathbf{A}'))}[l(\Gamma_\theta(\mathbf{X}, \mathbf{A}'), \mathbf{Y}') - l(\Gamma_\theta(\mathbf{X}, \mathbf{A}), \mathbf{Y}')]\right| \\
& + \left|\mathbb{E}_{\mathbf{X} \sim p(X), \mathbf{A} \sim p(A|E=E_{tr}), \mathbf{Y} \sim p(Y|A=\mathbf{A}), \mathbf{A}' \sim p(A|E=E_{te}), \mathbf{Y}' \sim p(Y|A=\mathbf{A}'))}[l(\Gamma_\theta(\mathbf{X}, \mathbf{A}), \mathbf{Y}') - l(\Gamma_\theta(\mathbf{X}, \mathbf{A}), \mathbf{Y})]\right| \\
\leq& \mathbb{E}_{\mathbf{X} \sim p(X), \mathbf{A} \sim p(A|E=E_{tr}), \mathbf{A}' \sim p(A|E=E_{te}), \mathbf{Y}' \sim p(Y|A=\mathbf{A}'))}\left[|l(\Gamma_\theta(\mathbf{X}, \mathbf{A}'), \mathbf{Y}') - l(\Gamma_\theta(\mathbf{X}, \mathbf{A}), \mathbf{Y}')|\right] \\
& + \mathbb{E}_{\mathbf{X} \sim p(X), \mathbf{A} \sim p(A|E=E_{tr}), \mathbf{Y} \sim p(Y|A=\mathbf{A}), \mathbf{A}' \sim p(A|E=E_{te}), \mathbf{Y}' \sim p(Y|A=\mathbf{A}'))}\left[|l(\Gamma_\theta(\mathbf{X}, \mathbf{A}), \mathbf{Y}') - l(\Gamma_\theta(\mathbf{X}, \mathbf{A}), \mathbf{Y})|\right].
\end{aligned}
\tag{24}
$$

Moreover, due to the Lipschitz continuity of $l$ and $\phi_{dec}$, we have

$$|l(\Gamma_\theta(\mathbf{X}, \mathbf{A}'), \mathbf{Y}') - l(\Gamma_\theta(\mathbf{X}, \mathbf{A}), \mathbf{Y}')| \leq L_1 \cdot \|\mathbf{Z}(T; \mathbf{A}') - \mathbf{Z}(T; \mathbf{A})\|_2, \tag{25}$$

$$|l(\Gamma_\theta(\mathbf{X}, \mathbf{A}), \mathbf{Y}') - l(\Gamma_\theta(\mathbf{X}, \mathbf{A}), \mathbf{Y})| \leq L_2 \cdot \|\mathbf{Y}' - \mathbf{Y}\|_2, \tag{26}$$

where $L_1$ and $L_2$ denote the Lipschitz constants. Combing Eqn. 25 and Eqn. 26 with Eqn. 24, we have

$$
\begin{aligned}
|\mathcal{R}(\Gamma_\theta; E_{te}) - \mathcal{R}(\Gamma_\theta; E_{tr})| \leq & L_1 \cdot \mathbb{E}_{\mathbf{A} \sim p(A|E_{tr}), \mathbf{A}' \sim p(A|E_{te})}\left[\|\mathbf{Z}(T; \mathbf{A}') - \mathbf{Z}(T; \mathbf{A})\|_2\right] \\
& + L_2 \cdot \mathbb{E}_{(\mathbf{A}, \mathbf{Y}) \sim p(A, Y|E_{tr}), (\mathbf{A}', \mathbf{Y}') \sim p(A, Y|E_{te})}\left[\|\mathbf{Y}' - \mathbf{Y}\|_2\right].
\end{aligned}
\tag{27}
$$

The conclusion for the main theorem can be obtained via combining Eqn. 27 and Eqn. 9 using the triangle inequality.

### B.2. Proof for Theorem 3.2

For the advective diffusion equation with the coupling matrix $\mathbf{C}$ pre-computed by attention network $\eta(\mathbf{z}_u(0), \mathbf{z}_v(0))$ and fixed velocity $\mathbf{V} = \mathbf{D}^{-1/2}\mathbf{A}\mathbf{D}^{-1/2}$, we have its closed-form solution

$$\mathbf{Z}(t) = e^{-(\mathbf{I} - \mathbf{C} - \beta\mathbf{V})t}\mathbf{Z}(0), \quad t \geq 0. \tag{28}$$

To prove the perturbation bound w.r.t. the change of graph structures, we introduce the following lemma.

**Lemma B.1.** *Let $\mathbf{X}, \mathbf{E} \in \mathbb{R}^{n \times n}$, and let $||\cdot||$ be a submultiplicative matrix norm. Suppose there exist constants $M \geq 1$, $\omega \geq 0$ such that for all $\mathbf{Y} \in \mathbb{R}^{n \times n}$, $||e^{\mathbf{Y}}|| \leq Me^{\omega\|\mathbf{Y}\|}$. Then the following perturbation bound holds:*

$$\|e^{\mathbf{X}+\mathbf{E}} - e^{\mathbf{X}}\| \leq \|\mathbf{E}\| \cdot M^2 \cdot e^{\omega(\|\mathbf{X}\|+\|\mathbf{E}\|)}. \tag{29}$$

*Proof.* Define path $\mathbf{X}(s) := \mathbf{X} + s\mathbf{E}$ for $s \in [0, 1]$. Then:

$$e^{\mathbf{X}+\mathbf{E}} - e^{\mathbf{X}} = \int_0^1 \frac{d}{ds} e^{\mathbf{X}(s)} \, ds. \tag{30}$$

Using the integral form of the derivative of the matrix exponential (Van Loan, 1977), we have:

$$\frac{d}{ds} e^{\mathbf{X}(s)} = \int_0^1 e^{(1-\theta)\mathbf{X}(s)} E e^{\theta\mathbf{X}(s)} \, d\theta. \tag{31}$$

Therefore:

$$e^{\mathbf{X}+\mathbf{E}} - e^{\mathbf{X}} = \int_0^1 \left( \int_0^1 e^{(1-\theta)\mathbf{X}(s)} E e^{\theta\mathbf{X}(s)} \, d\theta \right) ds. \tag{32}$$

Taking norms and applying submultiplicativity:

$$\|e^{\mathbf{X}+\mathbf{E}} - e^{\mathbf{X}}\| \leq \int_0^1 \int_0^1 \|e^{(1-\theta)\mathbf{X}(s)}\| \cdot \|\mathbf{E}\| \cdot \|e^{\theta\mathbf{X}(s)}\| \, d\theta ds. \tag{33}$$

Using the growth bound assumption:

$$\|e^{(1-\theta)\mathbf{X}(s)}\| \leq Me^{\omega(1-\theta)\|\mathbf{X}(s)\|}, \quad \|e^{\theta\mathbf{X}(s)}\| \leq Me^{\omega\theta\|\mathbf{X}(s)\|}. \tag{34}$$

Multiplying these we have:

$$\|e^{(1-\theta)\mathbf{X}(s)}\| \cdot \|e^{\theta\mathbf{X}(s)}\| \leq M^2 e^{\omega\|\mathbf{X}(s)\|}. \tag{35}$$

Note that $\|\mathbf{X}(s)\| = \|\mathbf{X} + s\mathbf{E}\| \leq \|\mathbf{X}\| + \|\mathbf{E}\|$, so:

$$\|e^{\mathbf{X}+\mathbf{E}} - e^{\mathbf{X}}\| \leq \|\mathbf{E}\| \cdot M^2 \cdot \int_0^1 \int_0^1 e^{\omega\|\mathbf{X}+s\mathbf{E}\|} \, d\theta ds \leq \|\mathbf{E}\| \cdot M^2 \cdot e^{\omega(\|\mathbf{X}\|+\|\mathbf{E}\|)}. \tag{36}$$

The existence of $M$ and $\omega$ is guaranteed for every consistent matrix norm (Van Loan, 1977) such as spectral norm $\|\cdot\|_2$ considered in our analysis.

$\square$

Let $\mathbf{L} = \mathbf{I} - \mathbf{C} - \beta\mathbf{V}$ and $\mathbf{L}' = \mathbf{I} - \mathbf{C}' - \beta\mathbf{V}'$. We then apply the above lemma that holds even if $\mathbf{L}$ and $\mathbf{L}^\top$ are not commutable:

$$\|e^{-\mathbf{L}'T} - e^{-\mathbf{L}T}\|_2 \leq M^2 T \cdot \|\mathbf{L}' - \mathbf{L}\|_2 \cdot e^{\omega T\|\mathbf{L}\|_2} \cdot e^{\omega T\|\mathbf{L}'-\mathbf{L}\|_2}. \tag{37}$$

For spectral norm $||\cdot||_2$ in our case, the above result holds for $M = 1$ and $\omega = 1$.

We next prove the bound for the last term of Eqn. 37 by construction. Notice that the initial states are given by the encoder MLP: $\mathbf{Z}(0) = \phi_{enc}(\mathbf{X})$. According to our data generation hypothesis in Fig. 2, we know that node embeddings are generated from the latents of each node (we use $\mathbf{u}_u$ to denote the realization of $U_u$), i.e., $\mathbf{x}_u = g(\mathbf{u}_u; W)$ and the graph adjacency is generated through a pair-wise function $a_{uv} = h(\mathbf{u}_u, \mathbf{u}_v; W, E)$. Since $g$ is injective, we assume $g^{-1}$ as its inverse mapping.

We define by $\eta \circ \phi_{enc}$ the function composition of $\eta$ and $\phi_{enc}$ that establishes a mapping from input node features $\mathbf{X}$ to the attention-based coupling $\mathbf{C}$. According to the universal approximation results that hold for MLPs on the compact set (Hornik et al., 1989), we can construct a mapping induced by $\eta \circ \phi_{enc}$ to obtain a propagation matrix in the form of $\mathbf{C} = \overline{\mathbf{C}} - (\beta + \epsilon)\mathbf{V}$, where $\overline{\mathbf{C}}$ is independent from $\mathbf{A}$ and $\epsilon > 0$ is an arbitrary small number. To be specific, the construction of the mapping can be achieved by $\eta \circ \phi_{enc} = m \circ h \circ g^{-1}$:

- $g^{-1}$ maps the input feature $\mathbf{x}_u$ to $\mathbf{u}_u$;

- $h$ maps $(\mathbf{u}_u, \mathbf{u}_v)$ to $a_{uv}$;

- $m$ maps $a_{uv}$ to $c_{uv}$, where $c_{uv}$ denotes the $(u, v)$-th entry of $\mathbf{C}$.

Then consider the difference of node representations under topological shifts and we have $\|(\mathbf{C}' + \beta\mathbf{V}') - (\mathbf{C} + \beta\mathbf{V})\|_2 = \epsilon \cdot O(\|\Delta\tilde{\mathbf{A}}\|_2)$. Since $\|\Delta\tilde{\mathbf{A}}\|_2$ is bounded, for any positive integer $m$, there exists $\epsilon > 0$ such that $\exp(\epsilon \cdot \|\Delta\tilde{\mathbf{A}}\|_2) \leq \|\Delta\tilde{\mathbf{A}}\|_2^m$. Therefore, we have the conclusion:

$$e^{\|\mathbf{L}' - \mathbf{L}\|_2} = e^{\|(\mathbf{C}' + \beta\mathbf{V}') - (\mathbf{C} + \beta\mathbf{V})\|_2} \leq O(\|\Delta\tilde{\mathbf{A}}\|_2^m). \tag{38}$$

The theorem can be concluded by combining the result of Eqn. 37.

### B.3. Proof for Corollary 3.3

The conclusion follows by combing the results of Theorem 3.1 and Theorem 3.2.

### B.4. Proof for Proposition 3.4

The diffusion equation with the constant coupling matrix $\mathbf{C}$ has a closed-form solution $\mathbf{Z}(t) = e^{-(\mathbf{I}-\mathbf{C})t}\mathbf{Z}(0)$, $t \geq 0$. To prove the proposition, we need to derive the bound of $\|e^{-(\mathbf{I}-\mathbf{C}')T} - e^{-(\mathbf{I}-\mathbf{C})T}\|_2$ for any $\mathbf{C}' \neq \mathbf{C}$. According to the result (3.5) of (Van Loan, 1977) we have

$$\|e^{-(\mathbf{I}-\mathbf{C}')T} - e^{-(\mathbf{I}-\mathbf{C})T}\|_2 \leq T\|\mathbf{C}' - \mathbf{C}\|_2\|e^{-(\mathbf{I}-\mathbf{C})T}\|_2 e^{\|(\mathbf{C}'-\mathbf{C})T\|_2}. \tag{39}$$

Given the fact $\mathbf{C}' - \mathbf{C} = \tilde{\mathbf{A}}' - \tilde{\mathbf{A}} = \Delta\tilde{\mathbf{A}}$, we have

$$\|e^{-(\mathbf{I}-\mathbf{C}')T} - e^{-(\mathbf{I}-\mathbf{C})T}\|_2 = O(\|\Delta\tilde{\mathbf{A}}\|_2 \exp(\|\Delta\tilde{\mathbf{A}}\|_2 T)). \tag{40}$$

This gives rise to the conclusion that

$$\|\mathbf{Z}(T; \mathbf{A}') - \mathbf{Z}(T; \mathbf{A})\|_2 = O(\|\Delta\tilde{\mathbf{A}}\|_2 \exp(\|\Delta\tilde{\mathbf{A}}\|_2 T)), \tag{41}$$

and we conclude the proof for the proposition.

### B.5. Proof for Corollary 3.5

By combing the results of Theorem 3.1 and Proposition 3.4, we have

$$
\begin{aligned}
\mathcal{D}_{ood-model}(\Gamma_\theta, E_{tr}, E_{te}) &= O(\mathbb{E}_{\mathbf{A}\sim p(A|E_{tr}), \mathbf{A}'\sim p(A|E_{te})}[\|\mathbf{Z}(T; \mathbf{A}') - \mathbf{Z}(T; \mathbf{A})\|_2]) \\
&\leq O(\mathbb{E}_{\mathbf{A}\sim p(A|E_{tr}), \mathbf{A}'\sim p(A|E_{te})}[\|\Delta\tilde{\mathbf{A}}\|_2 \exp(\|\Delta\tilde{\mathbf{A}}\|_2 T)]).
\end{aligned}
\tag{42}
$$

### B.6. Extension with Feature Transformations

The conclusion of Proposition 3.4 and Corollary 3.5 can be extended to the cases incorporating feature transformations and non-linear activation in-between propagation layers used in common GNNs, like GCN (Kipf & Welling, 2017). In particular, the diffusion model becomes the piece-wise diffusion equations with $K$ dynamics components as defined by Eqn. 18:

$$\frac{\partial \mathbf{Z}(t; k)}{\partial t} = (\mathbf{C} - \mathbf{I})\mathbf{Z}(t; k), \ \ t \in [t_{k-1}, t_k], \ \ \text{with initial conditions} \ \ \mathbf{Z}(t_{k-1}; k) = \phi_{int}^{(k)}(\mathbf{Z}(t_{k-1}; k-1)), \tag{43}$$

where $\phi_{int}^{(k)}$ denotes the node-wise feature transformation of the $k$-th layer. Based on this, can re-use the reasoning line of proofs for Proposition 3.4 to each component, and arrive at the exponential bound of node representation within the $k$-th dynamics:

$$\|\mathbf{Z}(t_k; \mathbf{A}', k) - \mathbf{Z}(t_k; \mathbf{A}, k)\|_2 = O(\|\Delta\tilde{\mathbf{A}}\|_2 \exp(\|\Delta\tilde{\mathbf{A}}\|_2(t_k - t_{k-1}))). \tag{44}$$

By stacking the results for each component, one can obtain the variation magnitude of the node representation yielded by the whole trajectory

$$\|\mathbf{Z}(T; \mathbf{A}') - \mathbf{Z}(T; \mathbf{A})\|_2 = O(\|\Delta\tilde{\mathbf{A}}\|_2 \exp(\|\Delta\tilde{\mathbf{A}}\|_2 T)). \tag{45}$$

## C. Approximation Strategies for Diffusion PDE Solutions

The closed-form solutions of linear diffusion equations often involve the form of matrix exponential $e^{-\mathbf{L}t}$, which is intractable for computing its exact value. There are many established techniques based on numerical approximations, e.g., series expansion, in this fundamental challenge. In our presented model in Sec. 4, we propose two implementation versions based on two approximation ways for handling the closed-form solution of the advective diffusion equations on graphs.

**Approximation with Linear Systems.** One scalable scheme proposed by (Gallopoulos & Saad, 1992) is via the extension of the minimax Padé-Chebyshev theory to rational fractions (Golub & Van Loan, 1989). This approximation technique has been utilized by (Patané, 2014) as an effective and efficient method for spectrum-free computation of the diffusion distances in 3D shape analysis. In specific, the matrix exponential of the form $e^{-\mathbf{L}t}$ is approximated by the combination of multiple matrix inverses:

$$\exp(-\mathbf{L}t) \approx -\sum_{i=1}^{r} \alpha_i(\mathbf{L} + \theta_i\mathbf{I})^{-1}, \tag{46}$$

where $\alpha_i$ and $\theta_i$ can be pre-defined parameters (Gallopoulos & Saad, 1992). To unleash the capacity of neural networks, in Sec. 4, our model implementation (ADVDIFFORMER-I) extends this scheme to a multi-head network where each head contributes to propagation with independently parameterized attention networks. The matrix inverse is computed with the linear system solver that is available in common deep learning tools (e.g., PyTorch) and supports automatic differentiation.

**Approximation with Geometric Series.** When the graph sizes become large, the matrix inverse can be computationally expensive. For better scalability, we can use the geometric series for approximation:

$$(\mathbf{L} + \theta_i\mathbf{I})^{-1} = \sum_{k=0}^{\infty} (-1)^k \theta_i^{-(k+1)} \mathbf{L}^k \approx \sum_{k=0}^{K} (-1)^k \theta_i^{-(k+1)} \mathbf{L}^k. \tag{47}$$

In this way, the matrix exponential can be approximately computed via a combination of finite series:

$$\exp(-\mathbf{L}t) \approx -\sum_{i=1}^{r} \alpha_i \sum_{k=0}^{K} (-1)^k \theta_i^{-(k+1)} \mathbf{L}^k. \tag{48}$$

In our model, the closed-form solution for the PDE induces $\mathbf{L} = (\mathbf{I} - \mathbf{C} - \beta\mathbf{V})$, and the summation in Eqn. 48 can be expressed as a weighted sum of $\mathbf{P}^k = (\mathbf{C} + \beta\mathbf{V})^k$ for $k = 0, \cdots, K$. Our model implementation (ADVDIFFORMER-S) proposed in Sec. 4 generalizes the weighted sum to a one-layer neural network.

## D. Model Implementations and Algorithms

In this section, we provide detailed and self-contained descriptions about our model architectures in Appendix D.1. Then in Appendix D.2, we discuss how to apply our model to various graph-structured data with additional input information. To make the presentation clear and focused on the model implementation side, we will re-define some notations that are originally defined in Sec. 4, where we formulate the model with the terminology of the PDE domain.

### D.1. Model Architectures

The model takes a graph $\mathcal{G} = (\mathcal{V}, \mathcal{E}, \mathbf{X}, \mathbf{A})$ as input, and output prediction in the downstream tasks. We assume the number of nodes in the graph $|\mathcal{V}| = N$, node feature matrix $\mathbf{X} \in \mathbb{R}^{N \times D}$ and graph adjacency matrix $\mathbf{A} \in \{0, 1\}^{N \times N}$. We use $\mathbf{D}$ to denote the diagonal degree matrix of $\mathbf{A}$. The normalized adjacency is denoted by $\tilde{\mathbf{A}} = \mathbf{D}^{-1/2}\mathbf{A}\mathbf{D}^{-1/2}$, and $\mathbf{1}$ is an all-one $N$-dimensional column vector. In this subsection, we assume $\mathcal{G}$ has no edge weight or edge feature for presentation, and with loss of generality, we will discuss how to incorporate these additional attributes in Appendix D.2.

D.1.1. INSTANTIATIONS AND PARAMETERIZATIONS

Our model is comprised of three modules: the encoder $\phi_{enc}$, the decoder $\phi_{dec}$, and the propagation network in-between the first two.

**Encoder:** The node features $\mathbf{X} = [\mathbf{x}_u]_{u \in \mathcal{V}} \in \mathbb{R}^{N \times D}$ are first mapped to embeddings in the latent space $\mathbf{Z}^{(0)} = [\mathbf{z}_u^{(0)}]_{u \in \mathcal{V}} \in \mathbb{R}^{N \times d}$ via the encoder: $\mathbf{Z}^{(0)} = \phi_{enc}(\mathbf{X})$. The encoder $\phi_{enc}(\cdot)$ is instantiated as a shallow MLP with non-linear activation (e.g., ReLU).

**Propagation:** The propagation network converts the initial node embeddings $\mathbf{Z}^{(0)}$ to the node representations $\mathbf{Z} = [\mathbf{z}_u]_{u \in \mathcal{V}} \in \mathbb{R}^{N \times d}$ (where $\mathbf{Z}^{(0)}$ and $\mathbf{Z}$ are the re-defined counterparts of $\mathbf{Z}(0)$ and $\mathbf{Z}(T)$, respectively, presented in Sec. 4). The propagation network is implemented via a multi-head network with $H$ heads involving the attention network $\eta^{(h)}(\cdot, \cdot)$ and feature transformation network $\phi_{FC}^{(h)}(\cdot)$. The latter is instantiated as a fully-connected layer $\mathbf{W}_{O,h}$, and the attention network is instantiated as a normalized dot-product positive similarity function:

$$
\eta^{(h)}(\mathbf{z}_u^{(0)}, \mathbf{z}_v^{(0)}) = 1 + \left( \frac{\mathbf{W}_{Q,h} \mathbf{z}_u^{(0)}}{\|\mathbf{W}_{Q,h} \mathbf{z}_u^{(0)}\|_2} \right)^\top \left( \frac{\mathbf{W}_{K,h} \mathbf{z}_v^{(0)}}{\|\mathbf{W}_{K,h} \mathbf{z}_v^{(0)}\|_2} \right),
$$

$$
\mathbf{C}_h = \{c_{uv}^{(h)}\}, \quad c_{uv}^{(h)} = \frac{\eta^{(h)}(\mathbf{z}_u^{(0)}, \mathbf{z}_v^{(0)})}{\sum_{w \in \mathcal{V}} \eta^{(h)}(\mathbf{z}_u^{(0)}, \mathbf{z}_w^{(0)})},
$$

(49)

where $\mathbf{W}_{Q,h} \in \mathbb{R}^{d \times d}$ and $\mathbf{W}_{K,h} \in \mathbb{R}^{d \times d}$ are trainable weights for query and key, respectively, of the $h$-th head. Then the node representations will be computed in different ways by two models.

- For ADVDIFFORMER-I, the node representations are calculated via

$$
\mathbf{L}_h = (1 + \theta)\mathbf{I} - \mathbf{C}_h - \beta \tilde{\mathbf{A}}, \quad h = 1, \cdots, H
$$
$$
\mathbf{Z}_h = \text{linsolver}(\mathbf{L}_h, \mathbf{Z}^{(0)}), \quad h = 1, \cdots, H
$$
$$
\mathbf{Z} = \sum_{h=1}^{H} \mathbf{Z}_h \mathbf{W}_{O,h},
$$

(50)

where $\mathbf{W}_{O,h} \in \mathbb{R}^{d \times d}$ is a trainable weight matrix. Alg. 1 summarizes the feed-forward computation of ADVDIFFORMER-I.

- For ADVDIFFORMER-S, the node representations are computed by

$$
\mathbf{P}_h = \mathbf{C}_h + \beta \tilde{\mathbf{A}}, \quad h = 1, \cdots, H
$$
$$
\mathbf{Z}_h^{(k)} = \mathbf{P}_h \mathbf{Z}_h^{(k-1)}, \quad k = 1, \cdots K, \quad h = 1, \cdots, H
$$
$$
\mathbf{Z} = \sum_{h=1}^{H} [\mathbf{Z}_h^{(0)}, \mathbf{Z}_h^{(1)}, \cdots, \mathbf{Z}_h^{(K)}] \mathbf{W}_{O,h},
$$

(51)

where $\mathbf{W}_{O,h} \in \mathbb{R}^{(K+1)d \times d}$ is a trainable weight matrix. To accelerate the computation of Eqn. 51, we can inherit the strategy used in (Wu et al., 2023) and alter the order of matrix products, which reduces the time and space complexity to $\mathcal{O}(N)$ (see Appendix D.1.2 for detailed illustration). Alg. 2 presents the feed-forward computation of ADVDIFFORMER-S that only requires $\mathcal{O}(N)$ algorithmic complexity.

**Decoder:** The decoder $\phi_{dec}(\cdot)$ transforms the node representations into prediction. Depending on the specific downstream tasks, the decoder can be implemented in different ways:

$$
\begin{aligned}
&\text{(node-level prediction): } \hat{y}_u = \text{MLP}(\mathbf{z}_u) \\
&\text{(graph-level prediction): } \hat{y} = \text{MLP}(\text{SumPooling}(\{\mathbf{z}_u\}_{u \in \mathcal{V}})) \\
&\text{(edge-level prediction): } \hat{y}_{uv} = \text{MLP}([\mathbf{z}_u, \mathbf{z}_v]).
\end{aligned}
$$

(52)

In particular, the softmax activation is used for output in classification tasks. For training, we adopt standard loss functions, i.e., cross-entropy for classification and mean square loss for regression.

---

**Algorithm 1** Feed-Forward of the Model ADVDIFFORMER-I.

---

**INPUT:** Node feature matrix $\mathbf{X}$ and normalized adjacency matrix $\tilde{\mathbf{A}}$.
$\mathbf{Z}^{(0)} = \phi_{enc}(\mathbf{X})$
**for** $h = 1, \cdots, H$ **do**

> $\mathbf{Z}_{Q,h} = \left[ \frac{\mathbf{W}_{Q,h}\mathbf{z}_u^{(0)}}{\|\mathbf{W}_{Q,h}\mathbf{z}_u^{(0)}\|_2} \right]_{u \in \mathcal{V}}, \quad \mathbf{Z}_{K,h} = \left[ \frac{\mathbf{W}_{K,h}\mathbf{z}_u^{(0)}}{\|\mathbf{W}_{K,h}\mathbf{z}_u^{(0)}\|_2} \right]_{u \in \mathcal{V}}$
> $\mathbf{U}_h = \mathbf{1}\mathbf{1}^\top + \mathbf{Z}_{Q,h}(\mathbf{Z}_{K,h})^\top$
> $\mathbf{C}_h = \text{diag}^{-1}(\mathbf{U}_h\mathbf{1})\,\mathbf{U}_h$
> $\mathbf{L}_h = (1+\theta)\mathbf{I} - \mathbf{S}_h - \beta\tilde{\mathbf{A}}$
> $\mathbf{Z}_h = \text{linsolver}(\mathbf{L}_h, \mathbf{Z})$

$\mathbf{Z} = \sum_{h=1}^H \mathbf{Z}_h \mathbf{W}_{O,h}$
**OUTPUT:** Node representations $\mathbf{Z}$ and predicted labels with $\phi_{dec}(\mathbf{Z})$.

---

**Algorithm 2** Feed-Forward of the Model ADVDIFFORMER-S (with $\mathcal{O}(N)$ complexity).

---

**INPUT:** Node feature matrix $\mathbf{X}$ and normalized adjacency matrix $\tilde{\mathbf{A}}$.
$\mathbf{Z}^{(0)} = \phi_{enc}(\mathbf{X})$
**for** $h = 1, \cdots, H$ **do**

> $\mathbf{Z}_{Q,h} = \left[ \frac{\mathbf{W}_{Q,h}\mathbf{z}_u^{(0)}}{\|\mathbf{W}_{Q,h}\mathbf{z}_u^{(0)}\|_2} \right]_{u \in \mathcal{V}}, \quad \mathbf{Z}_{K,h} = \left[ \frac{\mathbf{W}_{K,h}\mathbf{z}_u^{(0)}}{\|\mathbf{W}_{K,h}\mathbf{z}_u^{(0)}\|_2} \right]_{u \in \mathcal{V}}$
> $\mathbf{N}_h = \text{diag}^{-1}\left(N + \mathbf{Z}_{Q,h}\left((\mathbf{Z}_{K,h})^\top\mathbf{1}\right)\right)$
> $\mathbf{Z}_h^{(0)} = \mathbf{Z}^{(0)}$
> **for** $k = 1, \cdots, K$ **do**
>> $\mathbf{Z}_h^{(k)} = \mathbf{N}_h \cdot \left[ \mathbf{1}\left(\mathbf{1}^\top\mathbf{Z}_h^{(k-1)}\right) + \mathbf{Z}_{Q,h}\left((\mathbf{Z}_{K,h})^\top\mathbf{Z}_h^{(k-1)}\right) \right] + \beta\tilde{\mathbf{A}}\mathbf{Z}_h^{(k-1)}$
> $\mathbf{Z}_h = [\mathbf{Z}_h^{0)}, \mathbf{Z}_h^{(1)}, \cdots, \mathbf{Z}_h^{(K)}]$

$\mathbf{Z} = \sum_{h=1}^H \mathbf{Z}_h \mathbf{W}_{O,h}$
**OUTPUT:** Node representations $\mathbf{Z}$ and predicted labels with $\phi_{dec}(\mathbf{Z})$.

---

### D.1.2. ACCELERATION OF ADVDIFFORMER-S WITH LINEAR COMPLEXITY

We illustrate how to achieve the propagation of ADVDIFFORMER-S in Eqn. 51 with $\mathcal{O}(N)$ complexity. With the query and key matrices defined by $\mathbf{Z}_{Q,h} = \left[ \frac{\mathbf{W}_{Q,h}\mathbf{z}_u^{(0)}}{\|\mathbf{W}_{Q,h}\mathbf{z}_u^{(0)}\|_2} \right]_{u \in \mathcal{V}}$ and $\mathbf{Z}_{K,h} = \left[ \frac{\mathbf{W}_{K,h}\mathbf{z}_u^{(0)}}{\|\mathbf{W}_{K,h}\mathbf{z}_u^{(0)}\|_2} \right]_{u \in \mathcal{V}}$, the attention matrix $\mathbf{C}_h$ in Eqn. 49 is computed by (in the matrix form used for implementation)

$$\mathbf{C}_h = \text{diag}^{-1}\left(N + \mathbf{Z}_{Q,h}\left(\mathbf{Z}_{K,h}\right)^\top \mathbf{1}\right)\left(\mathbf{1}\mathbf{1}^\top + \mathbf{Z}_{Q,h}\left(\mathbf{Z}_{K,h}\right)^\top\right). \tag{53}$$

Computing the above result requires $\mathcal{O}(N^2)$ time and space complexity. Still, if we consider the feature propagation with $\mathbf{C}_h$, we have

$$\begin{aligned}
\mathbf{C}_h\mathbf{Z}_h^{(k)} &= \text{diag}^{-1}\left(N + \mathbf{Z}_{Q,h}\left(\mathbf{Z}_{K,h}\right)^\top \mathbf{1}\right) \cdot \left(\mathbf{1}\mathbf{1}^\top + \mathbf{Z}_{Q,h}\left(\mathbf{Z}_{K,h}\right)^\top\right) \cdot \mathbf{Z}_h^{(k)} \\
&= \text{diag}^{-1}\left(N + \mathbf{Z}_{Q,h}\left((\mathbf{Z}_{K,h})^\top\mathbf{1}\right)\right) \cdot \left[\mathbf{1}\left(\mathbf{1}^\top\mathbf{Z}_h^{(k)}\right) + \mathbf{Z}_{Q,h}\left((\mathbf{Z}_{K,h})^\top\mathbf{Z}_h^{(k)}\right)\right],
\end{aligned} \tag{54}$$

where the equality is achieved by altering the order of matrix products. The above computation only requires $\mathcal{O}(N)$ time and space complexity. The feed-forward computation of ADVDIFFORMER-S with $\mathcal{O}(N)$ acceleration is summarized in Alg. 2.

### D.2. Applicability of Our Model

In the main paper, we assume unweighted graphs without edge attribute features for model formulation. Without loss of generality, we next discuss how to extend our model to handle the edge weights and edge features.

**Edge Weights.** For weighted graphs, the adjacency matrix $\mathbf{A}$ would become a real matrix where the entry $a_{uv}$ denotes

the weight on the edge $(u, v) \in \mathcal{E}$. In this situation, we still have the corresponding normalized adjacency $\tilde{\mathbf{A}} = \mathbf{D}^{-1}\mathbf{A}$ or $\tilde{\mathbf{A}} = \mathbf{D}^{-1/2}\mathbf{A}\mathbf{D}^{-1/2}$, where $\mathbf{D} = \text{diag}([d_u]_{u \in \mathcal{V}})$ and $d_u = \sum_{v,(u,v)\in\mathcal{E}} a_{uv}$. Our model implementations can be trivially generalized to this case by using $\tilde{\mathbf{A}}$ as the propagation matrix for local message passing.

**Edge Features.** If the graph contains edge features, denoted by $\mathbf{E} = [\mathbf{e}_{uv}]_{(u,v)\in\mathcal{E}} \in \mathbb{R}^{|\mathcal{E}|\times D'}$, we introduce an encoding layer $\mathbf{W}_E \in \mathbb{R}^{D'\times d}$ for mapping the edge features into embeddings in the latent space and then incorporate them with node embeddings. In specific, we first compute the edge-to-node signals:

$$\mathbf{M} = [\mathbf{m}_u]_{u\in\mathcal{V}}, \quad \mathbf{m}_u = \sum_{v,(u,v)\in\mathcal{E}} \tilde{\mathbf{A}}_{u,v}\mathbf{W}_E\mathbf{e}_{uv}. \tag{55}$$

- For ADVDIFFORMER-I, we can modify Eqn. 50 as

$$\begin{aligned}
\mathbf{L}_h &= (1+\theta)\mathbf{I} - \mathbf{C}_h - \beta\tilde{\mathbf{A}}, \\
\mathbf{Z}_h &= \text{linsolver}\left(\mathbf{L}_h, (\mathbf{Z}^{(0)} + \mathbf{M})\right), \\
\mathbf{Z} &= \sum_{h=1}^{H} \mathbf{Z}_h\mathbf{W}_{O,h}.
\end{aligned} \tag{56}$$

- For ADVDIFFORMER-S, we can modify Eqn. 51 to be

$$\begin{aligned}
\mathbf{P}_h &= \mathbf{C}_h + \beta\tilde{\mathbf{A}}, \\
\mathbf{Z}^{(k)} &= \mathbf{P}_h(\mathbf{Z}^{(k-1)} + \mathbf{M}), \quad k = 1, \cdots K, \\
\mathbf{Z} &= \sum_{h=1}^{H} [\mathbf{Z}^{(0)}, \mathbf{Z}^{(1)}, \cdots, \mathbf{Z}^{(K)}]\mathbf{W}_{O,h}.
\end{aligned} \tag{57}$$

# E. Experiment Details

We supplement details for our experiments, regarding datasets, competitors, and implementations, for facilitating the reproducibility.

## E.1. Datasets

The datasets we use for the experiments in Sec. 5 span diverse domains and learning tasks. We summarize the statistics and brief descriptions for each dataset in Table 3, with the detailed information presented in the following subsections.

Table 3: Statistics and descriptions for experimental datasets.

| Dataset | #Nodes | #Edges | #Graphs | Train/Val/Test Split | Task | Metric |
|---|---|---|---|---|---|---|
| Synthetic-h | 1,000 | 14,064 - 32,066 | 12 | SBM (Homophily) | Node Regression | RMSE |
| Synthetic-d | 1,000 | 7,785 - 13,912 | 12 | SBM (Density) | Node Regression | RMSE |
| Synthetic-b | 1,000 | 14,073 - 59,936 | 12 | SBM (Block Number) | Node Regression | RMSE |
| Twitch | 1,912 - 9,498 | 31,299 - 153,138 | 7 | Geographic Domain | Node Classification | ROC-AUC |
| Arxiv | 169,343 | 1,166,243 | 1 | Publication Time | Node Classification | Accuracy |
| OGB-BACE | 10 - 97 | 10 - 101 | 1,513 | Molecular Scaffold | Graph Classification | ROC-AUC |
| OGB-SIDER | 1 - 492 | 0 - 505 | 1,427 | Molecular Scaffold | Graph Classification | ROC-AUC |
| DPPIN-nr | 143 - 5,003 | 22 - 25,924 | 12 | Protein Identification Method | Node Regression | RMSE |
| DPPIN-er | 143 - 5,003 | 22 - 25,924 | 12 | Protein Identification Method | Edge Regression | RMSE |
| DPPIN-lp | 143 - 5,003 | 22 - 25,924 | 12 | Protein Identification Method | Link Prediction | ROC-AUC |
| HAM | 8 - 25 | 7 - 29 | 1,987 | Relative Molecular Mass | Edge Classification | Accuracy |

### E.1.1. SYNTHETIC DATASETS

The synthetic datasets used in Sec. 5.1 simulate the graph data generation in Sec. 3.1, where the topological distribution shifts are caused by the difference of environments across training and testing data. In specific, we generate graphs of

$|\mathcal{V}| = 1000$ nodes, with the node features $\mathbf{X}$, graph adjacency matrix $\mathbf{A}$ and labels $\mathbf{Y}$ generated by the following process.

- Each node $u \in \mathcal{V}$ is assigned with a scalar $u_u$ randomly sampled from the uniform distribution $U[0,1]$.

- For the generation of node features $\mathbf{X} = [\mathbf{x}_u]_{u \in \mathcal{V}}$, we instantiate the node-wise function $g$ as a 2-layer MLP with ReLU activation and 4-dimensional output. Then the node feature $\mathbf{x}_u$ is generated through $\mathbf{x}_u = \text{MLP}(u_u)$.

- For the generation of graph adjacency $\mathbf{A} = [a_{uv}]_{u,v \in \mathcal{V}}$, we instantiate the pairwise function $h$ as the stochastic block model (Snijders & Nowicki, 1997) which generates edges according to the intra-block edge probability ($p_1$) and the inter-block edge probability ($p_2$). We map the nodes into $b$ blocks by the following rule: for node $u \in \mathcal{V}$, we assign it to the $k$-th block if $v_u \in [\frac{k-1}{b}, \frac{k}{b})$ (where $1 \le k \le b$). Then the edge $a_{uv}$ is randomly generated from a bernoulli distribution with $p_1$ if $u$ and $v$ are in the same block, and $p_2$ otherwise.

- For the generation of labels $\mathbf{Y}$, we consider the regression tasks and each node has a label $y_u$ generated through an ensemble model of a 2-layer GCN and a 1-layer DIFFormer (without using the graph-based propagation) with random initializations: $\mathbf{Y} = \text{gcn}(\mathbf{U}, \mathbf{A}) + \text{difformer}(\mathbf{U}, \mathbf{A})$, where $\mathbf{U} = [u_u]_{u \in \mathcal{V}}$.

Using the above data generation, we create 12 graphs with the indices #1$\sim$#12, and use the graph #1 for training, the graph #2 for validation, and the graphs #3$\sim$#12 for testing. The topological distribution shifts are introduced in three different ways as described in Sec. 5.1, where in each case, the detailed configurations for $p_1$, $p_2$ and $b$ are illustrated below.

- *Homophily Shift*: $p_1 = 0.1$, $b = 5$ and $p_2 = 0.01 + 0.05 * \frac{1}{12} * (i-1)$ for the graph #i.

- *Density Shift*: $b = 5$, $p_1 = 0.1 + 0.1 * \frac{1}{12} * (i-1)$ and $p_2 = 0.01 + 0.1 * \frac{1}{12} * (i-1)$ for the graph #i.

- *Block Shift*: $p_1 = 0.1$, $p_2 = 0.01$ and $b = 5 + (i-1)$ for the graph #i.

### E.1.2. INFORMATION NETWORKS

The citation network `Arxiv` provided by (Hu et al., 2020) consists of a single graph with 0.16M nodes, where each node represents a paper with the publication year (ranging from 1960 to 2020) and a subarea id (from 40 different subareas in total). The node attribute features are 128-dimensional obtained by averaging the word embeddings of the paper's title and abstract. The edges are given by the citation relationship between papers. The predictive task is to estimate the paper's subarea. We use the publication years to split the data: papers published before 2014 for training, within the range from 2014 to 2017 for validation, and on 2018/2019/2020 for testing. Since there is a single graph, to increase the difficulty of generalization, we consider the inductive setting: the testing nodes are not contained in the training graph. Table 5 demonstrates the dissimilar statistics for training/validation/testing graphs, manifesting the existence of topological shifts. Following the common practice, we use Accuracy as the evaluation metric.

Table 4: Statistics for training/validation/testing graphs on `Arxiv`. There is a single citation network that augments with time evolving, and with the data splits in the inductive setting, the previous graph is contained by the subsequent one.

|  | Train (1960-2014) | Valid (2015-2017) | Test 1 (2018) | Test 2 (2019) | Test 3 (2020) |
|---|---|---|---|---|---|
| # Target Nodes | 41,125 | 49,816 | 29,799 | 39,711 | 8,892 |
| # All Nodes | 41,125 | 90,941 | 120,740 | 160,451 | 169,343 |
| # All Edges | 102,316 | 374,839 | 622,466 | 1,061,197 | 1,166,243 |
| Max Degrees | 275 | 3,036 | 6,251 | 12,006 | 13,161 |
| Avg Degrees | 4.98 | 8.24 | 10.31 | 13.23 | 13.77 |

`Twitch` (Rozemberczki et al., 2021) is comprised of seven dis-connected graphs, where each node represents a Twitch user and edges indicate the friendship. Each graph is collected from the social network in a particular region, including DE, ENGB, ES, FR, PTBR, RU and TW. The node features are multi-hot with 2,545 dimensions indicating the user's profile. The predictive task is to classify the gender of the user. The seven networks with sizes ranging from 2K to 9K have distinct structural characteristics (such as densities and maximum degrees) as observed by (Wu et al., 2022a). We therefore split the data according to the geographic information: use the network DE for training, ENGB for validation, and the remaining networks for testing. The evaluation metric is ROC-AUC for binary classification.

### E.1.3. BIOLOGICAL PROTEIN INTERACTIONS

`DPPIN` (Fu & He, 2022) contains 12 individual dynamic network datasets at different scales, and each dataset is a dynamic protein-protein interaction network that describes the protein-level interactions of yeast cells. Each graph dataset is obtained by one protein identification method and consists of 36 graph snapshots, wherein each node denotes a protein that has a sequence of 1-dimensional continuous features with 36 time stamps. This records the evolution of gene expression values within metabolic cycles of yeast cells. The edges in the graph are determined by co-expressed protein pairs at one time, and each edge is associated with a co-expression correlation coefficient.

We consider the predictive tasks within each graph snapshot and ignore the temporal evolution between different snapshots. In specific, we use the graph topology of each snapshot as the observed graph adjacency $\mathbf{A}$ and use the gene expression values at the previous 10 time steps as node features $\mathbf{X}$. On top of this, we consider three different predictive tasks: 1) node regression for gene expression value at the current time (measured by RMSE); 2) edge regression for predicting the co-expression correlation coefficient (measured by RMSE); 3) link prediction for identifying co-expressed protein pairs (measured by ROC-AUC). Given the fact that each graph dataset has distinct sizes (ranging from 143 to 5,003 nodes) and distributions of 3-cliques and 4-cliques (ranging from 0 to hundreds) (Fu & He, 2022), we consider the dataset-level data splitting and use 6/1/5 graph datasets for training/validation/testing, which introduces topological distribution shifts.

### E.1.4. MOLECULAR MAPPING OPERATOR GENERATION

The *Human Annotated Mappings* (`HAM`) dataset (Li et al., 2020) consists of 1,206 molecules with expert annotated mapping operators, i.e., a representation of how atoms are grouped in a molecule. The latter segments the atoms of a molecule into groups of varying sizes. As an important step in molecular dynamics simulation, generating coarse-grained mapping operators aims to reproduce the mapping operators produced by experts. This task can be modeled as a graph segmentation problem (Li et al., 2020) which takes a molecule graph as input and outputs the labels for each edge that indicates if there is cut needed to partition the source and end atoms into different groups.

For data splits, we calculate the relative molecular mass of each molecule using the RDKit package[3], and rank the molecules with increasing mass. Then we use the first 70% molecules for training, the following 15% for validation, and the remaining for testing. This splitting protocol partitions molecules with different weights, and requires generalization from small molecules in the training set to larger molecules in the testing set.

Table 5: The range of relative molecular mass for training/validation/testing molecules in `HAM`.

| | Train | Valid | Test |
|---|---|---|---|
| Relative Molecular Mass | $108.18 \sim 273.34$ | $273.34 \sim 311.14$ | $311.14 \sim 762.94$ |

### E.2. Competitors

In our experiments, we compare with peer encoder backbones for graph learning tasks. The competitors span three aspects: 1) classical GNNs, 2) diffusion-based GNNs, and 3) graph Transformers. We briefly introduce the competitors and illuminate their connections with our model.

- **GCN** (Kipf & Welling, 2017) is a popular model that propagates node embeddings over observed graphs for computing node representations, which can be seen as the discretized version of graph diffusion equations with feature transformations.

- **GAT** (Velickovic et al., 2018) introduces attention networks for computing pairwise weights for neighboring nodes in the graph and propagates node signals with adaptive strengths given by the attention weights. GAT can be seen as the discretized version of the graph diffusion equation with time-dependent coupling matrices.

- **SGC** (Wu et al., 2019) proposes to simplify the GCN architecture by removing the feature transformations in-between propagation layers, reducing multi-layer propagation to one-layer. This brings up significant acceleration for training and inference. SGC can be seen as the discretization of the linear diffusion equation on graphs.

---

[3] https://github.com/rdkit/rdkit

- **GDC** (Klicpera et al., 2019) extends the graph convolution operator to graph diffusion convolution derived from the linear diffusion equation on graphs. We use its implementation version based on the heat kernel for diffusion coefficients.

- **GRAND** (Chamberlain et al., 2021a) proposes graph neural diffusion, a continuous PDE model, that generalizes manifold diffusion to graphs and then uses numerical schemes to solve the PDE. We compare with its linear version that implements the linear graph diffusion equation.

- **A-DGNs** (Gravina et al., 2023) is a stable graph neural architecture inspired by ODE on graphs that has provable capability to preserve long-range information between nodes and avoid gradient vanishing or explosion in training.

- **CDE** (Zhao et al., 2023) is a recently proposed continuous model derived from convection diffusion equations that is designed for addressing heterophilic graphs.

- **GraphTrans** (Wu et al., 2021) is a recently proposed Transformer for graph-structured data that satisfies the permutation-invariant property. The model architecture sequentially combines GNNs and Transformers in order, where the GNN can learn local, short-range structures and the Transformer can capture global, long-range relationships.

- **GraphGPS** (Rampásek et al., 2022) introduces a scalable and powerful Transformer model class for graph data and achieves state-of-the-art results on molecular property prediction benchmarks. We use its scalable implementation version with the Performer attentions (Choromanski et al., 2021).

- **DIFFormer** (Wu et al., 2023) is a scalable Transformer inspired by diffusion on graphs. The model is comprised of principled attention layers, which implements the diffusion iterations minimizing a global energy. The architecture integrates graph-based feature propagation and global attention in each layer. We use its version with simple diffusivity that only requires linear complexity and yields state-of-the-art results on some large-graph benchmarks.

### E.3. Implementation Details

**Computation Systems.** All the experiments are run on NVIDIA 3090 with 24GB memory. The environment is based on Ubuntu 18.04.6, Cuda 11.6, Pytorch 1.13.0 and Pytorch Geometric 2.1.0.

**Evaluation Protocol.** For all the experiments, we run the training and evaluation of each model with five independent trials, and report the mean and standard deviation results in our tables and figures. In each run, we train the model with a fixed budget of epochs and record the testing performance produced by the epoch where the model yields the best performance on validation data.

**Hyper-Parameters.** We use the grid search for hyper-parameter tuning on the validation dataset with the searching space described below.

- For information networks, hidden size $d \in \{32, 64, 128\}$, learning rate $\in \{0.0001, 0.001\}$, head number $H \in \{1, 2, 4\}$, the weight for local message passing $\beta \in \{0.2, 0.5, 0.8, 1.0\}$, and the order of propagation (only used for ADVDIFFORMER-S) $K \in \{1, 2, 4\}$.

- For molecular datasets, hidden size $d = 256$, learning rate $\in \{0.01, 0.005, 0.001, 0.0005, 0.0001, 0.00005\}$, dropout $\in \{0.0, 0.1, 0.3, 0.5\}$, head number $H \in \{1, 2, 4\}$, the weight for local message passing $\beta \in \{0.5, 0.75, 1.0\}$, the coefficient for identity matrix (only used for ADVDIFFORMER-I) $\theta \in \{0.5, 1.0\}$, and the order of propagation (only used for ADVDIFFORMER-S) $K \in \{1, 2, 3, 4\}$.

- For protein interaction networks, hidden size $d \in \{32, 64\}$, learning rate $\in \{0.01, 0.001, 0.0001\}$, head number $H \in \{1, 2, 4\}$, the weight for local message passing $\beta \in \{0.3, 0.5, 0.8, 1.0\}$, the coefficient for identity matrix (only used for ADVDIFFORMER-I) $\theta \in \{0.5, 1.0\}$, and the order of propagation (only used for ADVDIFFORMER-S) $K \in \{1, 2, 3, 4\}$.

## F. Additional Experimental Results

In this section, we supplement more experimental results including additional results for main experiments, ablation studies and hyper-parameter analysis.

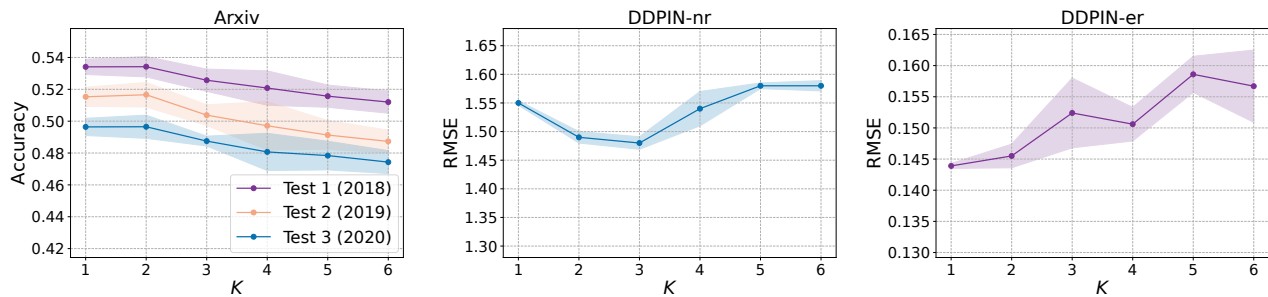

Figure 6: Model performance on `Arxiv` and `DPPIN` with different settings of $K$. The latter involves node regression (nr) and edge regression (er) tasks.

## F.1. Supplementary Results for Main Experiments

In Table 6, we present the ROC-AUC for each graph of `Twitch`. In Fig. 7 and 8, we show the generated results for more testing cases of molecular mapping operators in `HAM`.

Table 6: Result of ROC-AUC for each graph on `Twitch` where we use nodes in different networks to split the training, validation and testing data.

| | Train (DE) | Valid (ENGB) | Test 1 (ES) | Test 2 (FR) | Test 3 (PTBR) | Test 4 (RU) | Test 5 (TW) |
|---|---|---|---|---|---|---|---|
| **MLP** | $75.26 \pm 1.49$ | $63.48 \pm 0.15$ | $65.19 \pm 0.37$ | $62.25 \pm 0.28$ | $65.01 \pm 0.19$ | $54.92 \pm 0.33$ | $58.23 \pm 0.13$ |
| **GCN** | $69.55 \pm 0.34$ | $60.76 \pm 0.21$ | $63.75 \pm 0.44$ | $61.56 \pm 0.56$ | $63.26 \pm 0.42$ | $54.51 \pm 0.21$ | $55.72 \pm 0.28$ |
| **GAT** | $69.28 \pm 1.14$ | $59.80 \pm 0.42$ | $62.81 \pm 1.16$ | $60.65 \pm 0.92$ | $63.13 \pm 1.25$ | $53.80 \pm 0.27$ | $55.31 \pm 0.94$ |
| **SGC** | $71.68 \pm 0.33$ | $61.98 \pm 0.07$ | $65.12 \pm 0.15$ | $63.06 \pm 0.12$ | $64.14 \pm 0.19$ | $55.17 \pm 0.06$ | $56.83 \pm 0.20$ |
| **GDC** | $80.73 \pm 1.69$ | $62.14 \pm 0.30$ | $66.33 \pm 0.25$ | $60.70 \pm 0.51$ | $64.21 \pm 0.23$ | $56.60 \pm 0.24$ | $58.97 \pm 0.37$ |
| **GRAND** | $79.17 \pm 0.74$ | $62.48 \pm 0.39$ | $66.52 \pm 0.23$ | $61.62 \pm 0.62$ | $64.44 \pm 0.73$ | $56.42 \pm 0.38$ | $59.27 \pm 0.57$ |
| **A-DGNs** | $78.91 \pm 0.52$ | $61.52 \pm 0.34$ | $65.82 \pm 0.21$ | $60.59 \pm 0.56$ | $63.49 \pm 0.63$ | $55.74 \pm 0.32$ | $58.31 \pm 0.53$ |
| **CDE** | $80.21 \pm 0.35$ | $62.51 \pm 0.21$ | $65.62 \pm 0.17$ | $60.93 \pm 0.55$ | $63.92 \pm 0.57$ | $55.79 \pm 0.31$ | $58.42 \pm 0.42$ |
| **GraphTrans** | $79.17 \pm 0.74$ | $62.48 \pm 0.39$ | $66.52 \pm 0.23$ | $61.62 \pm 0.62$ | $64.44 \pm 0.73$ | $56.42 \pm 0.38$ | $59.27 \pm 0.57$ |
| **GraphGPS** | $74.49 \pm 1.35$ | $63.40 \pm 0.31$ | $66.85 \pm 0.32$ | $63.74 \pm 0.37$ | $65.03 \pm 0.58$ | $56.39 \pm 0.39$ | $58.63 \pm 0.83$ |
| **DIFFormer** | $73.12 \pm 0.52$ | $63.06 \pm 0.09$ | $66.68 \pm 0.15$ | $64.44 \pm 0.13$ | $65.23 \pm 0.20$ | $55.75 \pm 0.12$ | $58.91 \pm 0.37$ |
| **ADVDIFFORMER-S** | $75.46 \pm 0.28$ | $63.53 \pm 0.14$ | $66.78 \pm 0.14$ | $63.35 \pm 0.10$ | $65.68 \pm 0.06$ | $56.27 \pm 0.06$ | $60.48 \pm 0.21$ |

## F.2. Ablation Studies and Hyper-Parameter Analaysis

We next conduct more analysis on our proposed model by ablation studies on some key components and investigating the impact of hyper-parameters.

**Diffusion and Advection.** We conduct ablation studies on the advection term (i.e., the local message passing) and the diffusion term (i.e., the global attention). In Table 7 we report the results for ADVDIFFORMER-S on `Arxiv`, which shows that the two modules are indeed effective for producing superior generalization on node classification tasks.

Table 7: Ablation studies for ADVDIFFORMER-S on `Arxiv`.

| | Train (1960-2014) | Valid (2015-2017) | Test 1 (2018) | Test 2 (2019) | Test 3 (2020) |
|---|---|---|---|---|---|
| **ADVDIFFORMER** | $63.79 \pm 0.66$ | $55.25 \pm 0.14$ | $53.41 \pm 0.48$ | $51.53 \pm 0.60$ | $49.64 \pm 0.54$ |
| **ADVDIFFORMER** w/o diffusion | $64.65 \pm 1.10$ | $55.00 \pm 0.12$ | $52.45 \pm 0.27$ | $50.18 \pm 0.18$ | $48.01 \pm 0.24$ |
| **ADVDIFFORMER** w/o advection | $61.84 \pm 0.79$ | $54.31 \pm 0.24$ | $51.64 \pm 0.59$ | $49.65 \pm 0.53$ | $47.06 \pm 0.69$ |

**Impact of $K$.** The hyper-parameter $K$ (used for ADVDIFFORMER-S) controls the number of propagation orders in the model. In fact, the value of $K$ would impact how the structural information is utilized by the model. If $K$ is small, the model only utilizes the low-order structure, and large $K$ enables the usage of high-order structural information. Fig. 6 presents the model performance on `Arxiv` and `DPPIN` with $K$ ranging from 1 to 6. We observe that the optimal settings for $K$ are different across cases, and using larger $K$ can not necessarily yield better performance. This is because in these cases, the low-order structural information is informative enough for desired generalization.

**Impact of $\theta$.** Finally, we study the impact of $\theta$ used for computing $\mathbf{L}_h$ in ADVDIFFORMER-I. Table 8 shows the performance of ADVDIFFORMER-I on DPPIN with different $\theta$'s. We found that using $\theta$ close to 1 can bring up stably good performance, which is consistently manifested by experiments on other cases. Still, if $\theta$ is too small, e.g., close to 0, it would sometimes lead to numerical instability. This is due to that, in such a case, the matrix $\mathbf{L}_h$ could become a singular matrix.

Table 8: Testing accuracy of ADVDIFFORMER-I with different $\theta$'s in the edge regression task on DPPIN.

| $\theta$ | 0 | 0.5 | 1.0 | 2.0 |
|---|---|---|---|---|
| Accuracy | 0.241 | 0.154 | 0.147 | 0.149 |

## G. Current Limitations and Future Works

The generalization analysis in the current work focuses on the data-generating mechanism as described in Fig. 2 which is inspired and generalized by the random graph model. While this mechanism can in principle reflect real-world data generation process in various graph-structured data, in the open-world regime, there could exist situations involving topological distribution shifts by diverse factors or their combination. Future works can extend our framework for such cases where inter-dependent data is generated with different causal mechanisms. Another future research direction lies in the instantiation of the diffusion and advection operators in our model. Besides our choice of MPNN architecture to implement the advection process, other possibilities include structural and positional embeddings. We leave this line of exploration for the future, along with the analysis for the generalization capabilities of more general (e.g., non-linear) versions of the advective diffusion equation and other architectural choices.

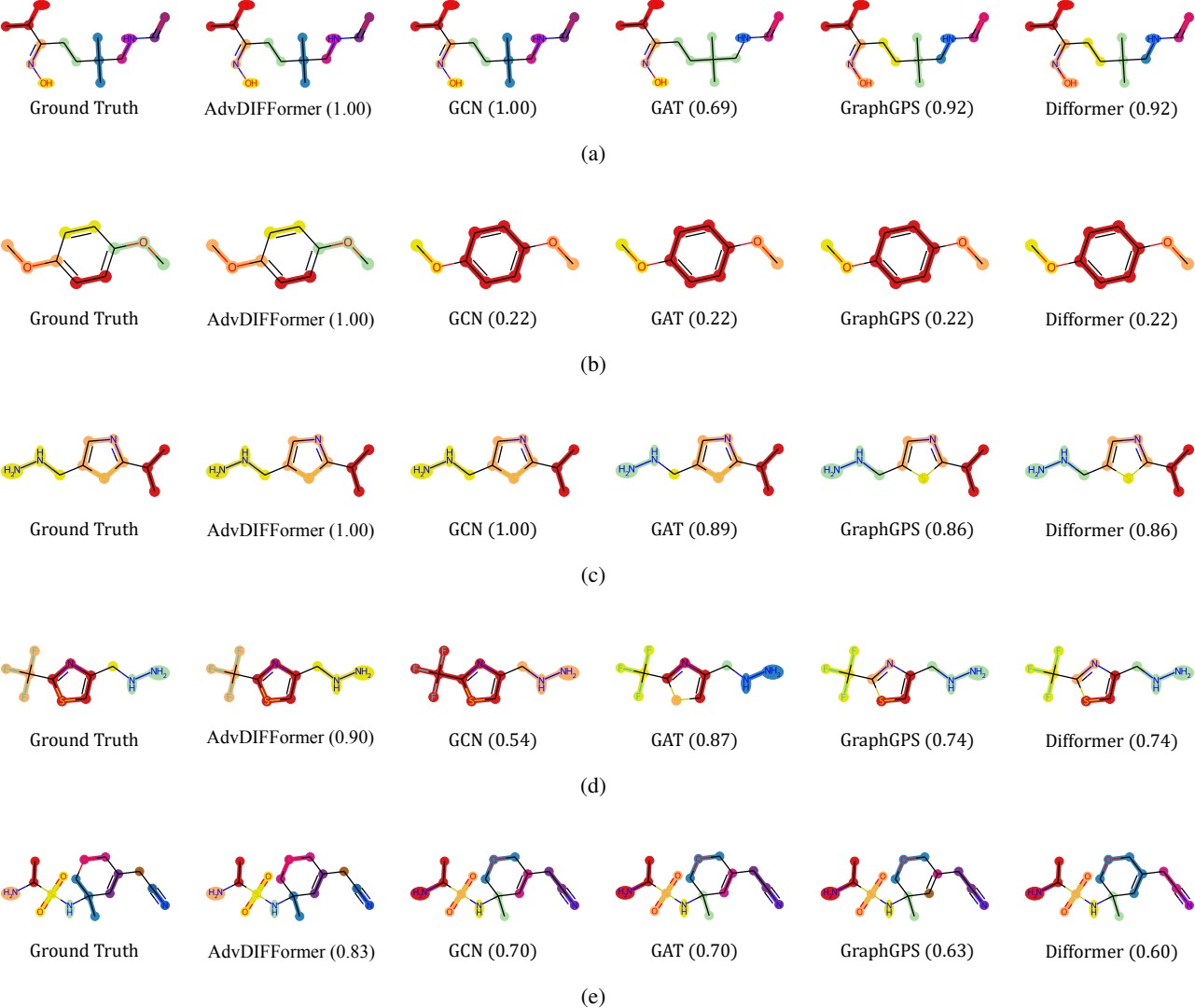

Figure 7: Additional testing cases for molecular mapping operators generated by different models and the expert annotations (ground-truth). For each case, we report the score (the higher is better) that measures the closeness between the generated results and the expert annotations.

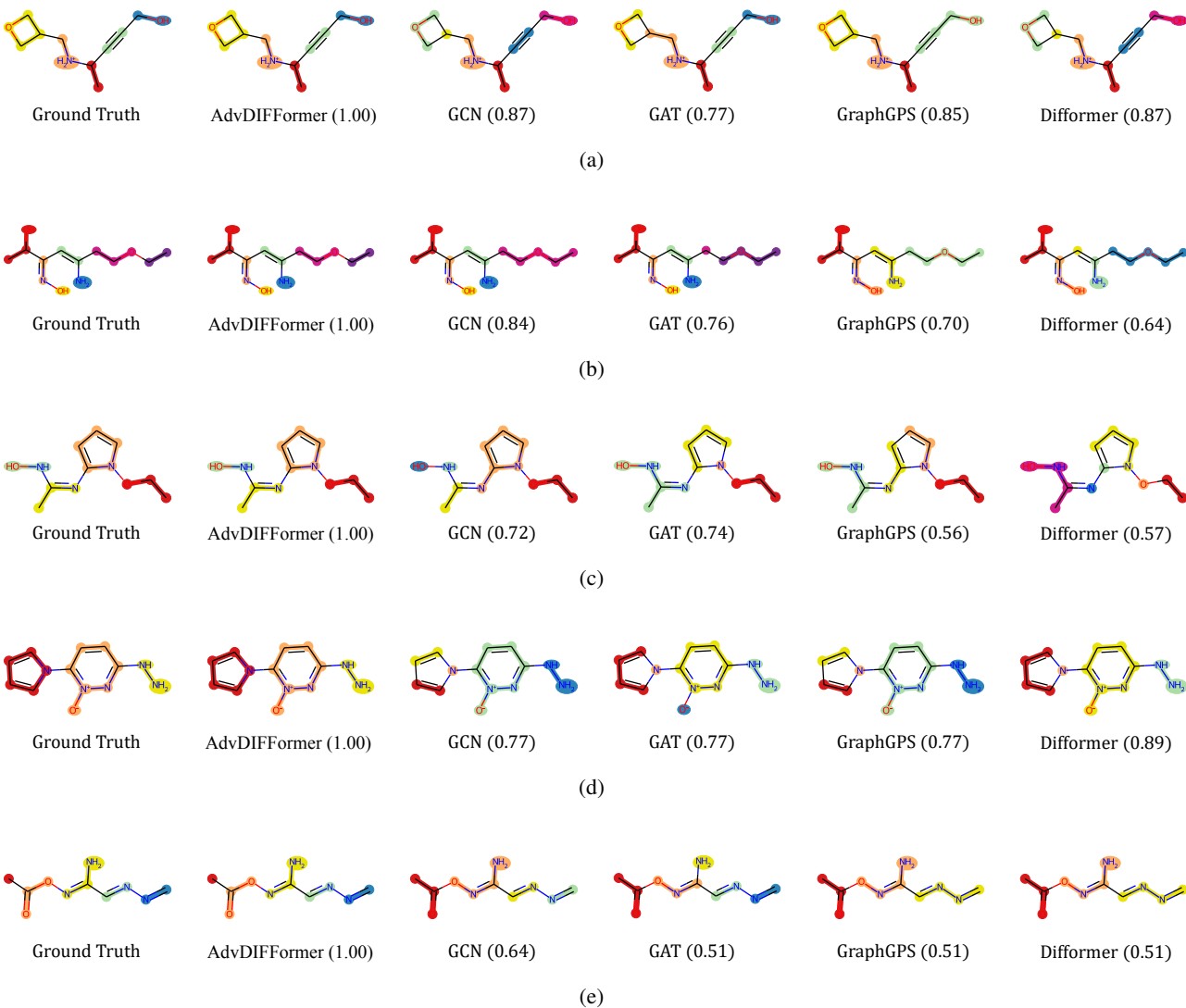

Figure 8: Additional testing cases for molecular mapping operators generated by different models and the expert annotations (ground-truth). For each case, we report the score (the higher is better) that measures the closeness between the generated results and the expert annotations.

