# OpenReview forum: "Supercharging Graph Transformers with Advective Diffusion"
_ICML.cc/2025/Conference — ICML 2025 poster_

### Official Review · Reviewer_sDvY · 2025-03-11

**Overall Recommendation:** 3

**Summary:**

The paper analysis the generalization capabilities of a specific class of diffusion-based graph models, namely those following the advective diffusion transformer, under topology distribution shifts. Specifically, it highlights that while the generalization gap for local diffusion—relying solely on the adjacency matrix—grows exponentially with changes in adjacency, incorporating a non-local propagation mechanism, in a transformer-like style, alongside the adjacency matrix reduces this gap to a polynomial scale. These findings are validated through experiments on a synthetic benchmark as well as several real-world datasets.

**Claims And Evidence:**

Most of the proofs presented in the paper are based on the result (3.5) of (Van Loan, 1977). That result requires the matrix to commute with its traspose. However, that is not necessary the case for the $I-C-\beta V$ matrix. In my opinion, additional assumptions needs to be done regarding the original graph topology and the coupling matrix C in order for the results to remain true. Since all the theoretical results presented in the paper (Theorem 3.2, Proposition 3.4 and everything derived from them) relies on this result, I would appreciate a clarification from the authors regarding the precise conditions under which their results hold. Specifically, could the authors clarify whether their framework implicitly assumes certain structural properties of C and the underlying graph, or whether alternative justifications exist for applying Van Loan’s result in this context?

**Essential References Not Discussed:**

N/A

**Experimental Designs Or Analyses:**

In Table 1 the paper reports out-of-memory results for the GraphTrans architecture. However, the proposed model presented in Section 4 and Algorithm 2 also relies on a fully connected graph, with the same type of attention mechanism as in the GraphTrans model. I am wondering why the GraphTrans memory requirement is higher than the one for AdiT-Series model presented in this paper.

**Methods And Evaluation Criteria:**

The setup presented in the paper, with a test split consisting of graphs with shifted topological distribution represent an important and under-explored  area in geometric deep learning. The experimental results are interesting and in line with the theoretical observations.

**Other Comments Or Suggestions:**

Section C in the appendix contains a missing reference (row 797, row 823, row 808, row 830)

**Other Strengths And Weaknesses:**

As mentioned above, my main concern regards the fault in the proofs. In my opinion, additional assumptions are made without being stated which can reduce the contribution of the work. If there is a way of bypassing them I am happy to raise my score to acceptance.

**Questions For Authors:**

Please see the boxes above.

**Relation To Broader Scientific Literature:**

The performance of graph neural networks under distribution shift remains an under-explored area within the graph community. In this regard, I believe the paper introduces theoretical novelty. As for the proposed model, its architecture can be seen as an interpolation between a local graph diffusion and a fully connected transformer, which somewhat diminishes its individual contribution. However, this generalization also offers advantages, as the theoretical results can be specialized to these two widely used standard architectures.

**Theoretical Claims:**

Please see "Claims And Evidence" section.

---

> ### Author Rebuttal · Authors · 2025-04-01
>
> Thank you for your time and thoughtful feedback.
>
> **Q1: Concerns about theoretical assumptions and (3.5) of (Van Loan, 1977)**
>
> We appreciate the opportunity to clarify this point. The reviewer is correct that the result (3.5) of (Van Loan, 1977) [1] requires the matrix to commute with its transpose. This condition **does hold for Prop 3.4**, as the normalized adjacency matrix $\tilde {\mathbf A} = \mathbf D^{-1/2}\mathbf A \mathbf D^{-1/2}$ is symmetric under the standard assumption that the observed graph is undirected—a common setting adopted by GNNs [e.g., 2–4] and justified via the graphon model described in Sec. 3.1. We’ll clarify this assumption in the revised paper.
>
> For Thm 3.2, we agree that the symmetry of $\mathbf I - \mathbf C - \beta \mathbf V$ may not generally hold, especially since $\mathbf C$ (attention matrix) may be asymmetric. While enforcing symmetry on $\mathbf C$ would technically resolve this, it would overly restrict the model’s design space. Instead, we provide an alternative proof that **bypasses the need for (3.5) of (Van Loan, 1977) entirely**:
>
> > **Lemma.** Let $X, E \in \mathbb{R}^{n \times n}$, and let $||\cdot||$ be a submultiplicative matrix norm. Suppose there exist constants $M \geq 1$, $\omega \geq 0$ such that for all $Y \in \mathbb{R}^{n \times n}$, $||e^Y|| \leq M e^{\omega ||Y||}$. Then the following perturbation bound holds:
> $$
> ||e^{X+E} - e^X|| \leq ||E|| \cdot M^2 \cdot e^{\omega (||X|| + ||E||)}.
> $$
> **Proof:** Define path $X(s) := X + sE$ for $s \in [0, 1]$. Then:
> $$
> e^{X+E} - e^X = \int_0^1 \frac{d}{ds} e^{X(s)} \ ds.
> $$
> Using the integral form of the derivative of the matrix exponential [5], we have:
> $$
> \frac{d}{ds} e^{X(s)} = \int_0^1 e^{(1 - \theta)X(s)} E e^{\theta X(s)} \ d\theta.
> $$
> Therefore:
> $$
> e^{X+E} - e^X = \int_0^1 \left( \int_0^1 e^{(1 - \theta)X(s)} E e^{\theta X(s)} \ d\theta \right) ds.
> $$
> Taking norms and applying submultiplicativity:
> $$
> ||e^{X+E} - e^X|| \leq \int_0^1 \int_0^1 ||e^{(1 - \theta)X(s)}|| \cdot ||E|| \cdot ||e^{\theta X(s)}|| \ d\theta ds.
> $$
> Using the growth bound assumption:
> $$
> ||e^{(1 - \theta)X(s)}|| \leq M e^{\omega (1 - \theta)||X(s)||}, \quad
> ||e^{\theta X(s)}|| \leq M e^{\omega \theta ||X(s)||}.
> $$
> Multiplying these we have:
> $$
> ||e^{(1 - \theta)X(s)}|| \cdot ||e^{\theta X(s)}|| \leq M^2 e^{\omega ||X(s)||}.
> $$
> Note that $||X(s) || = ||X + sE|| \leq ||X|| + ||E||$, so:
> $$
> ||e^{X+E} - e^X|| \leq ||E|| \cdot M^2 \cdot \int_0^1 \int_0^1 e^{\omega ||X + sE||} d\theta ds \leq ||E|| \cdot M^2 \cdot e^{\omega(||X|| + ||E||)}.
> $$
>
> The existence of $M$ and $\omega$ is guaranteed for every consistent matrix norm [5] such as spectral norm $||\cdot||_2$ considered in our analysis.
>
> Now consider the proof for our Thm 3.2, let $\mathbf L = \mathbf I - \mathbf C - \beta \mathbf V$ and $\mathbf L' = \mathbf I - \mathbf C' - \beta \mathbf V'$. We can apply the above lemma even if $\mathbf L$ and $\mathbf L^\top$ are not commutable:
> $$
> ||e^{-\mathbf L' T} - e^{-\mathbf LT}||_2 \leq M^2 T \cdot ||\mathbf L' - \mathbf L||_2 \cdot e^{\omega T ||\mathbf L||_2 } \cdot e^{\omega T ||\mathbf L' - \mathbf L||_2  }.
> $$
> For spectral norm $||\cdot||_2$ considered in our analysis, the above result holds for $M=1$ and $\omega=1$. Also, the last term can be bounded using the same argument of Line 727-747 in our proof which leads to
> $$
> e^{||\mathbf L' - \mathbf L||_2} = e^{||(\mathbf C' + \beta \mathbf V') - (\mathbf C + \beta \mathbf V)||_2} \leq O(||\Delta \tilde{\mathbf A}||_2^m).
> $$
> Therefore, $||e^{-\mathbf L' T} - e^{-\mathbf LT}||_2$ is bounded with polynomial order w.r.t. $||\Delta \tilde{\mathbf A}||_2$ and Thm 4.2 can be concluded.
>
> We will revise the paper to replace the use of (3.5) of (Van Loan, 1977) in Thm. 3.2 with this updated, assumption-free proof.
>
> **Q2: GraphTrans vs. ADiT-Series memory cost**
>
> GraphTrans uses the original global attention mechanism, resulting in quadratic time and memory complexity w.r.t. node number. This makes it infeasible for large graphs like Arxiv (0.2M nodes), leading to the out-of-memory issue in Table 1.
>
> In contrast, ADiT-Series employs a scalable global attention that preserves all-pair interactions while reducing time and memory complexity to linearity w.r.t. node number. This enables our model to handle large graphs efficiently. Details of this acceleration are presented in Appendix D.1.2, and we’ll highlight this distinction more clearly in the revision.
>
> **Q3: Missing references in Appendix**
>
> Thank you for catching these. We will fix these typos in the revised paper.
>
> [1] Van Loan, C. The sensitivity of the matrix exponential.
>
> [2] GRAND: graph neural diffusion, ICML 2021
>
> [3] Diffusion improves graph learning, NeurIPS 2019
>
> [4]  Semi-supervised classification with graph convolutional networks, ICLR 2017
>
> [5] Higham et al. Functions of Matrices: Theory and Computation. SIAM, 2008.
>
> Please let us know if further clarification is needed. Thank you again!

---

> > ### Comment · Reviewer_sDvY · 2025-04-03
> >
> > Thank you for the detailed reply. I am happy with the revised proof.

---

> > > ### Author Response · Authors · 2025-04-03
> > >
> > > Thanks for your nice feedback and we are glad that our rebuttal addresses your concern. We'll modify the paper accordingly when we have the chance for revision.
> > >
> > > It would be greatly appreciated if you can re-consider the rating on our work. Thanks again for your time and consideration.

---

### Official Review · Reviewer_tsoh · 2025-03-13

**Overall Recommendation:** 3

**Summary:**

This paper proposes a framework called Advective Diffusion Transformer (ADIT) to address topological distribution shifts in graph-structured data. By formulating a continuous diffusion equation augmented with an advection term, the authors effectively combine non-local propagation with local message passing. They justify ADIT via a theoretical analysis showing that the model can control and limit the impact of topological changes on performance. Empirically, ADIT outperforms standard graph neural networks and transformer-based baselines across synthetic, informational (citation/social), and molecular/protein datasets.

**Claims And Evidence:**

The authors claim that ADIT maintains stable performance under distribution shifts in graph structure and the paper presents bounds indicating that ADIT's embeddings respond in a polynomially bounded way to changes in adjacency. However, the authors primarily focus on carefully controlled synthetic scenarios and specific real-world splits. In practice, shifts may be more diverse (e.g., entirely new nodes, partial re-labeling, dynamic edges). It remains unclear whether the same polynomial-bounded embedding behavior would hold under more chaotic or evolving graph processes.
Although the theoretical discussion is convincing in isolated PDE contexts, real data can deviate from the carefully posited assumptions (e.g., injectivity of the latent generator). Thus, additional evidence or bounding under more relaxed conditions could strengthen the claim.

**Essential References Not Discussed:**

N/A

**Experimental Designs Or Analyses:**

The topological splits for real datasets (by year or region) do not fully guarantee that all topological changes (e.g., changes in the node sets, or partial re-labeling) are tested. Additional experiments varying the node set or combining multiple shift factors might reveal more nuanced results.

**Methods And Evaluation Criteria:**

ADIT solves a PDE that unifies local and global propagation. Two variants approximate the continuous-time solution. However, there are some weaknesses:
First, the computational overhead of matrix-inversion-based or large K-step expansions could be significant on huge graphs. While the authors propose optimizations (e.g., multi-head parallelism, Chebyshev expansions), the paper does not thoroughly compare memory/time profiles across different scales or under real-time constraints.
Second, the weighting hyperparameter is introduced to balance advection vs. diffusion, but the method to select it is somewhat heuristic.
For the evaluation criteria, it remains uncertain how these chosen splits map onto the PDE-based assumptions.

**Other Comments Or Suggestions:**

No other comments.

**Other Strengths And Weaknesses:**

Other strengths:
ADIT offers a clear PDE-based argument for combining local and global message passing; this is theoretically principled and practically relevant.
Other weaknesses:
For extremely large graphs, it is unclear if the matrix exponential expansions remain viable or how approximations degrade performance.
The model introduces several PDE-inspired hyperparameters. A systematic procedure for selecting them under real constraints or partial label availability is not thoroughly explored.

**Questions For Authors:**

No other questions.

**Relation To Broader Scientific Literature:**

Recent research (e.g., robust GNNs under adversarial attacks or continual learning over evolving graphs) also addresses shifting topologies from different perspectives. A more explicit comparison or synergy with these areas (e.g., adversarially perturbed edges) would highlight whether ADIT can handle worst-case shifts or only mild, distributional ones.
The authors stress the PDE analogy but do not benchmark or contrast directly with non-PDE-based robust or domain-adversarial GNN approaches. Further references and direct comparisons might clarify whether PDE-based methods inherently outperform, or if domain-invariant approaches could match ADIT’s stability.

**Theoretical Claims:**

The authors present bounding arguments for out-of-distribution generalization error, contrasting polynomial vs. exponential sensitivity to adjacency changes. However, the authors assume “injectivity” in the latent generator g; real data might see degenerate or repeated node embeddings, which can weaken the claims. Moreover, the PDE analysis hinges on classical results that require smoothness assumptions and well-posedness of the underlying continuous system. Graph data can be highly irregular, with discrete or non-smooth connectivity. Some justification of how classical PDE frameworks carry over to more irregular or large-scale networks would be valuable.

---

> ### Author Rebuttal · Authors · 2025-04-01
>
> Thank you for your thoughtful review and constructive feedback.
>
> **Q1: Theoretical assumptions and justification on more relaxed real-world conditions**
>
> Our PDE model indeed serves as a simplified abstraction, but it does not introduce new assumptions beyond the standard setting of graph representation learning [1–4]. Despite the simplification, PDE-based analysis allows for understanding GNN behavior, even on irregular and large-scale graphs, as shown in prior work [1-4] and our own.
>
> The injectivity assumption in Thm 3.2 is indeed nontrivial but reasonable: since $g$ maps from a low-dimensional latent space to a higher-dimensional observation space, collisions (i.e., degenerate mappings) are rare in practice. Removing this assumption would be interesting, but it introduces substantial technical challenges that we'd like to leave to future work.
>
> Crucially, our model performs well on real datasets that do not satisfy the theoretical assumptions, confirming that the PDE framework yields effective architectures even under relaxed real-world conditions. While we agree that shifts in the wild can be more complex, our goal is to establish a principled framework and demonstrate its robustness to a wide range of topological shifts—including more diverse settings added below.
>
> **Q2: Adding experiments with more complex distribution shifts (combination of multiple shift factors, new nodes/edges and partial label availability)**
>
> Before presenting new results, we kindly note that Table 1 already included dynamic shifts where new nodes and edges appear over time. We adopt an inductive setting (see details in Appendix E.1.2) where test nodes and their edges are excluded during training. Apart from this, we add experiments on a new dataset OGBL-COLLAB that is also a dynamic graph where new nodes/edges are added to the test graph. See our rebuttal to **Q1** of Reviewer ZJpC for results and details.
>
> To further strengthen evaluation, we add a more challenging variant of the Arxiv dataset combining multiple shift types (node/edge changes and partial label availability). On top of the data splits that already involve dynamic shifts, we mask 10 of 40 classes for training data, and evaluate on test data with full labels. As shown below, while all models degrade under this setting, ADiT-Series maintains clear superiority:
>
> ||GCN|GAT|GraphGPS|DIFFormer|ADiT-Series
> |-|-|-|-|-|-|
> |**Test Acc ($\uparrow$) with single shift**|46.46±0.85|46.50±0.21|46.46±0.95|44.30±2.02|49.64±0.54|
> |**Test Acc ($\uparrow$) with multiple shifts**|39.29±0.91|38.82±0.52|39.46±1.33|37.30±1.14|44.64±0.82|
>
> **Q3: Scalability to large graphs and comparison of computational costs**
>
> ADiT-Series scales linearly with graph size, enabling it to scale up to large graphs. In practice, we use small values of
> $K$ (e.g., 1–4), yielding strong performance and efficiency. We add comparison of computational costs on the Arxiv graph (0.2M nodes). ADiT-Series consumes moderate GPU memory and much fewer time costs than Transformer competitors (DIFFormer and GraphGPS):
>
> ||GCN|GAT|GraphGPS|DIFFormer|ADiT-Series
> |-|-|-|-|-|-|
> |**Train Time per Epoch (second)**|7.66|18.12|198.32|73.99|25.21|
> |**Train GPU Memory (GB)**|2.4|10.8|13.7|5.3|4.2|
>
> **Q4: Systematic procedure for hyperparameter selection**
>
> We provided systematic studies for all hyper-parameters introduced by our model (such as $\beta$, $\theta$ and $K$) in Sec. 5.2 and Appendix F.2, with searching procedures and spaces detailed in Appendix E.3.
>
> **Q5: Comparison with non-PDE-based robust or domain-adversarial GNNs**
>
> We appreciate the suggestion that can increase our impact to broader area of robust graph learning. We add a comparison on the DPPIN edge regression task against SR-GNN [5] and EERM [6], two robust GNNs—EERM uses domain-adversarial training. While all models show similar average RMSE, our models substantially improve worst-case performance, highlighting better stability under distribution shifts:
>
> ||SR-GNN|EERM|ADiT-Inverse|ADiT-Series
> |-|-|-|-|-|
> |**Test Average RMSE ($\downarrow$)**|0.170±0.003|0.172±0.006|0.166±0.006|0.167±0.004|
> |**Test Worst RMSE ($\downarrow$)**|0.201±0.013|0.207±0.018|0.184±0.011|0.188±0.010|
>
> This suggests that PDE-inspired architectures can offer robustness comparable to domain-invariant GNNs, without requiring adversarial training schemes.
>
> [1] GRAND: graph neural diffusion, ICML 2021
>
> [2] GRAND++: graph neural diffusion with a source term, ICLR 2022
>
> [3] Gradient gating for deep multi-rate learning on graphs, ICLR 2023
>
> [4] Understanding convolution on graphs via energies, TMLR 2024
>
> [5] Shift-robust gnns: Overcoming the limitations of localized graph training data, NeurIPS 2021
>
> [6] Handling Distribution Shifts on Graphs: An Invariance Perspective, ICLR 2022
>
> Please let us know if further clarification is needed. Thank you again!

---

> > ### Comment · Reviewer_tsoh · 2025-04-04
> >
> > Thanks the authors for the efforts in providing the rebuttals. Most of my concerns are well targeted. I will raise my score.

---

> > > ### Author Response · Authors · 2025-04-04
> > >
> > > Thanks for your nice feedback. We are glad that our rebuttal addresses your concerns. We'll revise the paper accordingly and incorporate the new results when we have the chance for revision. Thank you again for your time and constructive comments.

---

### Official Review · Reviewer_ZJpC · 2025-03-13

**Overall Recommendation:** 3

**Summary:**

This paper proposes advective diffusion transformer, a model that provably controls generalization error with topological shifts. The model has been evaluated on synthetic and several real-world datasets that verify its superiority compared with existing local and non-local graph neural networks/transformers.

**Claims And Evidence:**

The claims are clear with convincing evidence.

**Essential References Not Discussed:**

N/A

**Experimental Designs Or Analyses:**

The experimental designs are valid and rigor.

**Methods And Evaluation Criteria:**

The method and its evaluation is technically sound.

**Other Comments Or Suggestions:**

N/A

**Other Strengths And Weaknesses:**

Nice unified view of global and local message passing.

**Questions For Authors:**

1. How is the performance of the proposed approach on the OGB benchmarks and OGB-LSC PCQM4Mv2 dataset?

2. Are there any results on the benchmarks from [1]?

[1] Dwivedi et al. Benchmarking graph neural networks.

**Relation To Broader Scientific Literature:**

A general graph diffusion framework with competitive empirical performance, though not a fully novel approach since there are a series of existing works on graph neural diffusion.

**Theoretical Claims:**

The claims seem to be sound though I did not carefully check the proof.

---

> ### Author Rebuttal · Authors · 2025-04-01
>
> Thank you for your time in reviewing our paper and for the constructive feedback.
>
> **Q1: "How is the performance of the proposed approach on the OGB benchmarks?"**
>
> We appreciate the suggestion to include additional datasets. We added results on OGBL-COLLAB, a link prediction task suited for evaluating models under topological shifts. We follow the temporal splits from [2], which naturally introduce distribution shifts. We found that our model (ADiT-Series) achieves substantial improvements over GNN/Transformer baselines:
>
> | | GCN  | GAT | GraphGPS | DIFFormer | ADiT-Series
> | - | - | - | - | - | - |
> | **Test Hit@50 ($\uparrow$)** | 50.42±1.13 | 51.50±0.96 | 53.98±0.98 | 53.24±0.42 | 56.24±0.75 |
>
>
> **Q2: "Are there any results on the benchmarks from [1]?"**
>
> We also added results on Wiki-CS from [1]. As the original split uses random sampling (i.e., no distribution shift), we introduce a harder setting: nodes are sorted by degree and split into 25%/25%/50% for train/val/test. This allows us to evaluate performance under topological shifts. ADiT-Series matches or slightly outperforms baselines under the original split, and shows notable gains in the harder setting with topological shifts:
>
> | | GCN | GAT  | GraphGPS | DIFFormer | ADiT-Series
> | - | - | - |  - |  - |  - |
> | **Test Acc ($\uparrow$) with original split** | 77.46±0.85 | 76.90±0.82 | 78.34±0.88 | 79.39±0.62 | 79.53±0.42
> | **Test Acc ($\uparrow$) with harder split** | 61.98±0.63 | 63.52±0.40 | 64.21±0.98 | 65.34±1.03 | 68.21±0.79
>
> **Q3: "Not a fully novel approach since there are a series of existing works on graph neural diffusion"**
>
> We agree that graph neural diffusion has been studied extensively. However, our work introduces novelty in two key aspects: (1) we address topological shifts, a challenging but underexplored setting where training and test graphs differ; and (2) our approach is derived from a continuous PDE formulation that unifies local and non-local propagation, in contrast to prior works that focus primarily on local diffusion and static structures.
>
> [1] Dwivedi et al. Benchmarking graph neural networks.
>
> [2] Hu et al. Open graph benchmark: Datasets for machine learning on graphs, NeurIPS 2020.
>
> Please let us know if further clarification is needed. Thank you again!

---

### Decision · Program_Chairs · 2025-05-01

**Decision:**

Accept (poster)

**Comment:**

The paper studies generalization under topological shifts and proposes a model motivated by the advective diffusion equation. They provide both theoretical justification and empirical evaluation of the resulting model. Reviewers raised concerns about some of the theoretical analysis and requested more extensive evaluation. The authors clarified both points in the rebuttal, leading to one score being raised, and thus all reviewers rating the paper "weak accept".